# Bangladesh's vulnerability to cyclonic coastal flooding

Aurélia Bernard[1,2,6], Nathalie Long[1], Mélanie Becker[1], Jamal Khan[3,4], and Sylvie Fanchette[5]

[1]LIENSs, La Rochelle Université - CNRS, 2 rue Olympe de Gouges, 17000 La Rochelle, France.
[2]CERG-C Coordination, Faculty of Earth Sciences, University of Geneva - Rue des Maraîchers 13, 1205 Geneva, Switzerland.
[3]LEGOS, Université de Toulouse - CNRS - IRD - UPS - CNES,14 avenue Edouard Belin 31400 Toulouse, France.
[4]IWFM/BUET, Dhaka, Bangladesh.
[5]CESSMA, IRD, Paris Diderot University, 8 Paul Ricœur Square, 75013 Paris, France.
[6]ESPACE, Avignon Université, CNRS, Campus Hannah Arendt, 74 rue Louis Pasteur, 84000 Avignon, France.

*Correspondence to:* Nathalie Long (nathalie.long@univ-lr.fr)

**Abstract.** In the Ganges-Brahmaputra-Meghna delta, covering most of Bangladesh, more than 165 million people live in low-lying coasts facing major extreme climatic events, such as cyclones. This article reviews the current scientific literature publications (2007-2020) in order to define vulnerability in the context of coastal Bangladesh facing cyclonic flooding. Based on this review, a new metric, called 'socio-spatial vulnerability index' (SSVI), is defined as function of both the probability of the cyclonic flood hazard and the sensitivity of delta inhabitants. The main result shows that the districts of Shariatpur, Chandpur and Barisal situated in the tidal floodplain of the Ganges-Brahmaputra-Meghna delta, are in the fourth quartile, i.e. highest category, the most vulnerable areas. These districts are very densely populated (from 870 up to 1400 inhabitants/km$^2$) and exposed to inundation hazards host a large number of vulnerability factors. Finally, the delta's mouth was identified as a very vulnerable area to cyclonic flooding as well.

## 1 Introduction

Tropical cyclones are among the most devastating extreme natural events. When a cyclone approaches coast, it can induce an increase of sea level, called storm surge. This storm surge can cause severe flooding of vast coastal low-lying areas and leads to profound societal and economic impacts. This is especially the case for the regions surrounded by the semi-enclosed sea of the Bay of Bengal. Although the frequency of the cyclone is weaker than in Atlantic and Pacific Oceans, most of them make landfall (Balaguru et al., 2014). This, combined with low-lying and high-populated areas, causes catastrophic impacts (Hoque et al., 2019). In April 1991, for example, 15 million people were affected in Bangladesh by the flooding that followed the cyclone Gorky landfall, causing about 140'000 fatalities and 1.8 billion of dollars' worth of damage (source: International Emergency Events Database: EM-DAT: www.emdat.be). These cyclone-induced storm surges are more frequent during pre-monsoon (May) and post-monsoon (October–November) season, when the flow at Ganges-Brahmaputra river system is relatively low and the rainfall moderate (Uddin et al. 2019b), thereby reducing the likelihood of experiencing compound events, i.e. combined with other flood types as pluvial and fluvial (Haque et al. 2018). Concerning the contribution of the heavy rainfall from landfalling cyclones to the floods, additional to the storm surge, it still remains unexplored for this region. Therefore, hereafter, the study focused only on the cyclone-induced storm surge flood hazard. The development of accurate and understandable cyclone flooding knowledge and tools is therefore crucial to define effective strategies for adaptation to these extreme events. Today, the cyclone flood risk assessments have become an essential worldwide tool to assist in setting appropriate protection measures. However, the

multiplication of the cyclone flood risk assessments raises important questions about how their findings are comparable, generalizable and easily transferable to the decision-makers and communities.

In this study, we wonder, first and foremost, about the risk terminology. Although this terminology lacks uniformity between the social and the natural sciences, as well as within each one (Roberts et al., 2009), both approaches recognize that risk combines hazard, vulnerability and exposure (and sometimes coping capacities and resilience). While natural sciences emphasize hazard above all and consider vulnerability mainly from the element at risk in physical and technical viewpoints, social sciences rather focus on structural factors that reduce the human system capacity to cope with a range of hazards (Baum et al., 2009; Roberts et al., 2009). Vulnerability has many facets influencing each other and may also be measured by susceptibility to loss, on physical (natural and man-made environments), social, economic, institutional and systemic levels. Therefore, more than the exposure or the hazard, the vulnerability is a key factor for the relevance of all cyclone flood risk assessments.

In order to address the vulnerability to cyclone flooding, current approaches (among other: Ghosh et al., 2019; Hoque et al., 2019; Quader et al., 2017; Sahoo and Bhaskaran, 2018) tend to focus on flood hazard and the other social vulnerability patterns, but are rarely integrated into an interdisciplinary framework considering interaction between both (Rabby et al., 2019). Previous studies define the vulnerability of places, also called physical vulnerability, following if the risk of flooding can be enhanced or reduced by changes into the natural and built environment (Cutter, 2003; England and Knox, 2016). For cyclone flooding, topography, dikes, embankments, roads, houses and land use can therefore influence the physical vulnerability. Moreover, although these places might have the same probability to be flooded by cyclone, the sensitivity of people and their capacity to cope with flood will be different (O'Hare and White, 2018). Therefore, in this work we considered a cyclone flood event in this integrated and interdisciplinary framework, as the intersection between physical vulnerability and individual/group sensitivity, resulting into the concept of 'socio-spatial vulnerability' to cyclone flood (Forrest et al., 2020).

Thus, more than developing a new vulnerability concept, the novelty of our study lies in the methodological adaptation of existing approaches, like social vulnerability index (Flanagan et al., 2011), to the specific cyclone flooding context. In this way, we define a new metric, called 'socio-spatial vulnerability index' (SSVI), as function of both the probability of the cyclone flood hazard and the sensitivity of inhabitants. This metric is used to identify the areas where the largest number of the most vulnerable people are exposed to frequent cyclone flooding.

This study explores the potential for cyclone flood SSVI to highlight socio-spatial inequalities between the coastal districts in the case of Bangladesh, considered as one of the most vulnerable country in the world regarding extreme climatic events. In this region, 165 million people live on the low-lying land of the Ganges-Brahmaputra-Meghna (GBM) delta and face, among other things, catastrophic storm surge floods caused by cyclones (Ali, 1999; Becker et al., 2020; Nicholls et al., 2018; Uddin et al., 2019a). Like mentioned above, several studies have already been carried out to assess the cyclone flood vulnerability of Bangladesh coastal areas. However, as mentioned by Hoque et al. (2019), only two studies have proposed the production of a multi-parameter index as a tool to characterize the physical vulnerability in the central and western part of the GBM delta (Islam et al., 2016; Islam et al., 2015). Another work made by Mahmood et al. (2020) propose a cyclone flood vulnerability assessment located on the

mouth area of the GBM delta, which is based on the coastal vulnerability index developed by Hoque et al. (2019). Only physical parameters such as topography, shoreline change, and coastal slope (among others) were taken into account. Therefore, here also the social dimension of cyclone flood vulnerability was not assessed.

The paper is organized as follows: section 2 gives an overview of current paradigms used to conceptualize vulnerability; section 3 presents the region study; section 4 provides insights from a literature review into how vulnerability to cyclone flooding in Bangladesh is questioned and according to which indicators; and section 5 proposes a socio-spatial index for standardizing vulnerability to cyclone flooding in Bangladesh. Finally, two sections are devoted to the discussion and conclusion.

## 2 Overview about the vulnerability concept and its indicators

In the following, we provide a necessary overview of different vulnerability paradigms in order to understand how they were defined in these past decades, and how such a conceptualization can be achieved. Following the authors, vulnerability is seen either as static or dynamic, including exposure or not. Chambers (1989) initiated a formal definition as "exposure to contingencies and stresses and the difficulty which some communities experience while coping with such contingencies and stresses". Hence, he identified two binaries of vulnerability: an external (the exposure to external shocks) and an internal vulnerability (the incapacity to cope). Lastly, he suggested that it is determined by the management and accessibility of assets. Here, exposure is included but vulnerability is mainly considered as a state (Roberts et al., 2009).

The pressure and release model proposed by Blaikie (1994) considers that vulnerability is socially produced (Menoni et al., 2012). It is defined as: "the characteristics of a person or group in terms of their capacity to anticipate, cope with, resist, and recover from the impact of a natural hazard''. It focuses on both the social and temporal dimensions of a disaster (Kuhlicke et al., 2011) and proposes that vulnerability can be reduced through people's access to resources (Menoni et al., 2012).

Further developed by Wisner et al. (2004), the model includes a sense of progression into the vulnerability, which gradually evolves according to: root causes (the unequal distribution of power and assets), dynamic pressures (the society characteristics, such as population density and growth) and unsafe conditions (situations evolving as its expression when the hazard occurs). This model conceptualizes vulnerability as a process, but it underestimates the exposure in the risk equation (Roberts et al., 2009). Vulnerability as a dynamic process was additionally developed by Birkmann, (2006) in the BBC model which incorporates coping capacity into the vulnerability item of the risk equation. It also distinguishes between its social, economic and environmental components (Romieu et al., 2010). However, his model does not integrate physical vulnerability as impacting the social aspects. In most of these models, it is emphasized how it is a dynamic concept that does not only include socio-demographic characteristics.

Vulnerability is also the results of human interactions and networks and could change across time. De Marchi and Scolobig (2012) claim that the vulnerability of social systems is derived from characteristics of individuals, groups, but also from institutions. For those reasons, later authors offered to characterize its various components according

to its associated people and threatened assets. In addition, those authors adopt a systemic perspective in which they distinguish between the individual and the institutional components of social vulnerability in order to explore their interactions. Individual vulnerability stresses the perceptions and attitudes impeding people from preparing and coping sufficiently in case of disaster. However, institutional vulnerability implies that agencies and services could paradoxically weaken the overall capacity of a specific system to deal effectively with disasters (De Marchi and Scolobig, 2012). For example, the authors demonstrate how the presence of structural devices (barriers, dams and embankments) can lessen the perception of risk and mislead to a "symbol of complete safety". This institutional vulnerability leads to a situation where the bigger the defense system is, the greater the impression of invulnerability increases. This effect is well known in the literature as the "safety paradox" (Burby, 2006; Ferdous et al., 2020): increasing levels of flood protection can generate a sense of complacency among the protected people, which can reduce preparedness, thereby increasing vulnerability.

Furthermore, to assess vulnerability level, classical indicators are often social inequality indicators, such as: *age, gender, household composition, employment, income, educational level, ethnicity, race* (Blaikie, 1994; IPCC AR5, 2014). Other socio-economic indicators are added by the United Nations Organisation and the World Bank, such as: *gross domestic product (GDP) per inhabitant, human poverty index (HPI), inflation rate, population characteristics (density, growth, life expectancy at birth, illiteracy rate)*. These classical indicators of vulnerability may lack explanatory power and so this type of parameters using quantitative data should be discussed and put into perspective with qualitative indicators (Kuhlicke et al., 2011). Indicators relying only on census data lack its full understanding because they do not take into account people's perceptions and culture of risk, norms, traditions, beliefs and behaviors. Its other social dimensions, like "the ability of individuals and communities to cope seems to depend also on some intangible aspects, which can be described by terms such as energy, vigor, vitality, strength, courage, nerve, fortitude, and their antonyms" (De Marchi and Scolobig, 2012). Those aspects shape people's vulnerability, together with personal, contextual and material conditions.

As mentioned above, according to disciplinary approaches, the physical and social dimensions must be studied together and fit within a territory. This spatial dimension then appears unavoidable to assess vulnerability (Mazumdar and Paul, 2018). It enables, from a local to a national level, to define specific adaptation or mitigation strategies for each region, through the identification of hot spots, and to use the map as means of communication to meet the decision-makers' needs. As mentioned by Mazumdar and Paul (2018), vulnerability is specific according to population, factors and locations. This is why we have chosen to assess the vulnerability to cyclone flooding through a socio-spatial vulnerability index (SSVI), a tool that enables both quantifying and mapping its levels. Socio-spatial vulnerability takes into account social conditions and measures the resistance or resilience of populations to a hazard; at the same time, it integrates the potential exposures of these same populations in particular territories (Cutter et al., 2003; England and Knox, 2016; Forrest et al., 2020; O'Hare and White, 2018). Thus, we consider the vulnerability in a structural perspective (Hufschmidt, 2011) and where "exposure", i.e. the spatial intersection of flood hazard and population density, is included as one of its components.

**3 Region study**

Bangladesh is situated between India, Myanmar and the Bay of Bengal. This country is one of the most vulnerable countries in the world regarding extreme climatic events. The entirety of coastal Bangladesh is under 10 meters' elevation (Becker et al., 2019) and is highly populated with a mean of 1'240 people per km$^2$ in 2018 (World Bank, 2018), which increases its exposure to coastal floods. Though people of the deltas have learnt how to live with periodic coastal submersion, they face numerous constraints like ground instability because of fluvial alluvium and subsidence, insalubrity of swamps, and ground salinity, which compelled the population to adapt its cultural activities like planting new rice varieties (Ali, 1999) or shifting from rice cultivation to shrimp farming.

Conventional methods for reducing the effect of cyclonic flooding in this region are: embankments, polderization, coastal afforestation and shelter construction (Rahman et al. 2015). Since 1960s, Bangladesh has a widespread system of embankments and polders (land surrounded by embankments), which initially improve agriculture and water level management (Brown et al., 2018). However, as these earthen embankments systems were built for tidal flooding - without considering storm surges – they were not able to protect from extreme cases of cyclonic flooding. Besides, due to a lack of maintenance, most of coastal embankments nowadays are partially or completely eroded (Younus and Sharna, 2014). In 1972, a Cyclone Preparedness Program (CPP) was created by the Bangladesh's government to prepare weather forecasts and disaster warnings (Paul, 2009), whereby cyclone shelters were erected as multi-storey buildings – for example, in the form of "School cum Cyclone Shelter", to promote education as well as serve as shelter during storm-surges. Flood hazards challenge the high and growing population density, where the majority of urban Bangladeshis live in the main cities (Dhaka, Khulna and Chittagong; Fig. 1.a), which all contend with flood hazards. Floods threaten people's lives and assets, but can also damage livelihoods, agriculture and impact almost all socio-economic activities. Storm surge flooding in Bangladesh resulted in the greatest global death toll over the last century, such as the Cyclone Gorky in 1991, causing about 140'000 fatalities (Nicholls et al., 2018). In the previous decade, the floods triggered by cyclones Sdir in 2007 and Aila in 2009 caused thousands of fatalities (Ahmed et al., 2016; Paul, 2009) and displaced many others. In order to reduce people affected by cyclonic coastal flooding, a number of risk assessment studies (among other : Mallick et al. 2017; Quader et al. 2017; Hoque et al. 2019) have been conducted in this region to determine which actions should be prioritized for policy makers to implement.

Recent progress in hydrodynamic modeling for the GBM delta showed that to accurately reproduce patterns of cyclonic coastal flooding, several parameters are essential as accurate bathymetry and topography, inclusions of dikes, well-tuned bottom roughness, and forcing the model with accurate representation of the cyclonic wind and pressure fields (Khan et al., 2020; Krien et al., 2017). Using this robust hydrodynamical modelling schema, Khan et al. (2019) provided probabilistic cyclonic flood hazard maps over coastal GBM from a dataset of statistically and physically consistent synthetic cyclone events (Emanuel et al., 2006). We presented in Figure 1.a the cyclonic flooding level with a 50-year return-period estimated from Khan et al. (2019). This means that at a particular location in the GBM, the reported flooding height, due to cyclonic flooding, is expected to be observed on average once in 50 years. This type of metric aims to anticipate the cyclone flood occurrence and its consequences in order to alert the inhabitants and put in place effective strategies of evacuation. In dark blue on Figure 1.a, one observes the places which are exposed to flooding with the most extreme heights, up to eight meters from the mean sea level at the junction between the Ganges and the Meghna rivers. All the coastal islands areas are exposed to a 1-

in-50-year flood, though the biggest ones, such as Hatiya, Sandwip, and Kutubdia islands are not totally exposed to the cyclonic inundation on their total area. For these large coastal islands, the presence of dykes makes a very large gradient in the inundation hazard. Along almost all the coastline, blue colors are darker - excepting the South of Laxmipur district - meaning that the embankments and their polders behind are strongly exposed to the inundation. This directly threatens the houses settled by the dykes as authors underlined (Alam and Collins, 2010).

In addition to lots of places exposed to flooding with the most extreme heights, the population is aggregated around the main cities of the South: Khulna, Barisal, Chittagong and Cox's Bazar, as well as on big islands such as Sandwip and Kutubdia island (Fig. 1b). While the area in the southwestern coast, behind the Sundarbans forest, is not very populated compared with other parts of the delta, the districts of the central and east coasts are much denser, such as in the Chandpur Sadar upazila (sub-district) at the mouth of the GBM delta. In Khulna, the water height is lower but still significant when crossed with the high population density. In Shatkhira and Khulna, the west coastal districts, population density is lower but as the people of this area is highly nature resources-dependent, the flooding of the Sundarbans increases the vulnerability of the people in the area.

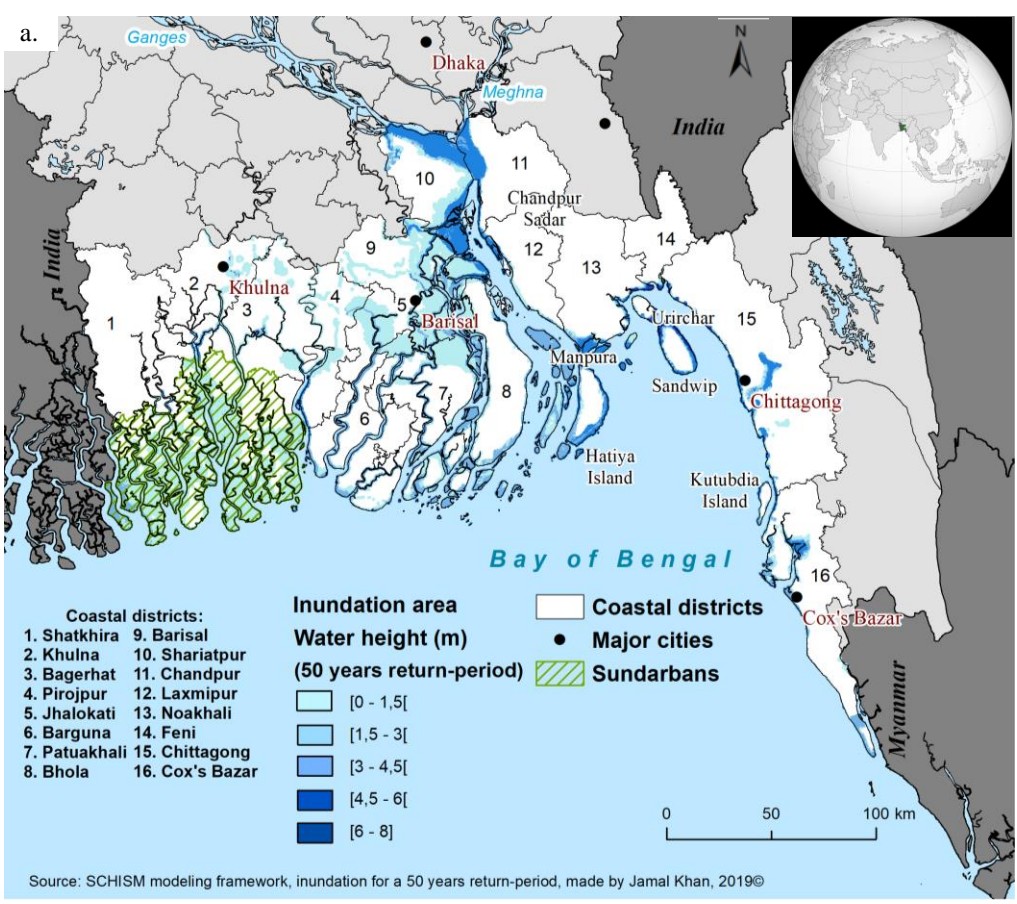

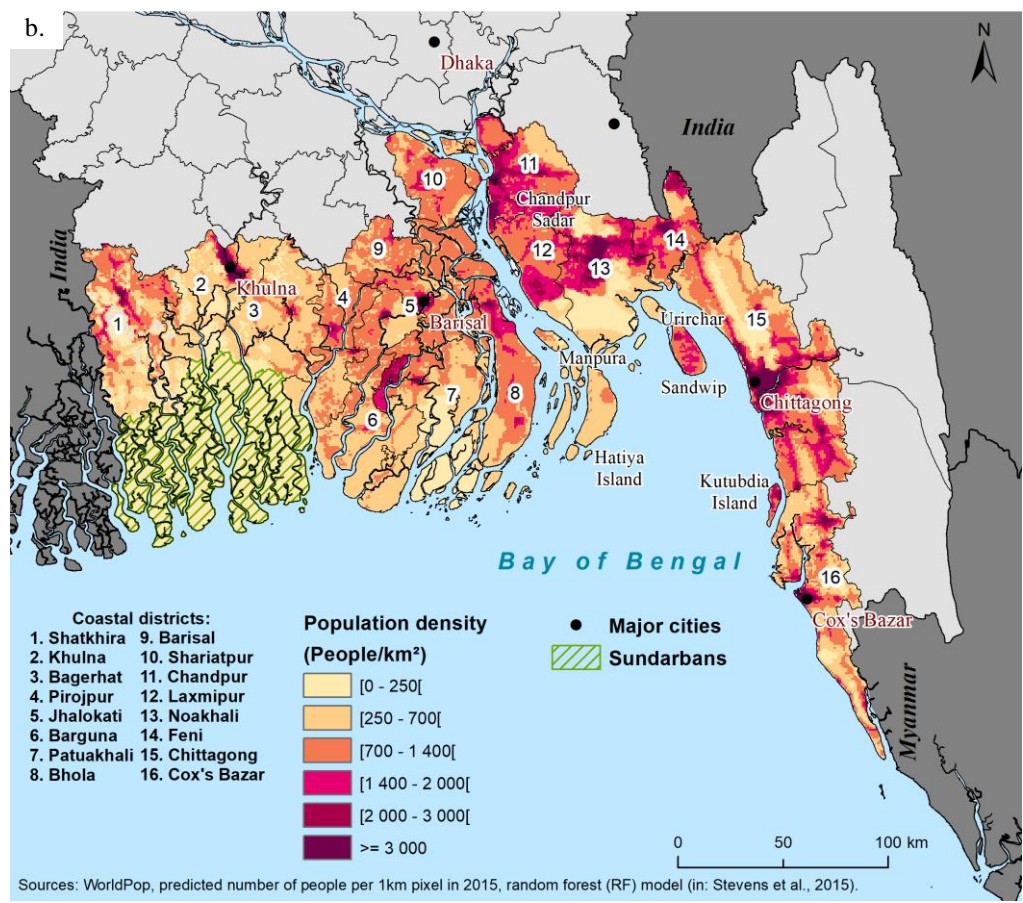

**Figure 1: Study area : a) cyclonic inundation level with a 50-year return-period estimated from a large number of hydrodynamic simulations (Khan et al., 2019). This means that at a particular location, the reported flooding height, due to cyclonic surges, is expected to be observed once in 50 years. b) population density in coastal districts of Bangladesh.**

**4 Assessing vulnerability to cyclonic coastal flooding in Bangladesh from a literature review**

**4.1 Methodology**

To define and assess the vulnerability to cyclonic coastal flooding in Bangladesh, a literature review was conducted. This review was limited to peer-reviewed articles published between 2007 and December 2020. The analysis period started in 2007, taking this year as a turning point in terms of developments of vulnerability frameworks. Indeed, in 2007, the Fourth Assessment Report of the Intergovernmental Panel on Climate Change (IPCC) set the significance of vulnerability social factors over the international research community. This review excludes non-peer-reviewed articles, in the aim to obtain a state of the art's review of the current knowledge extracted from scientific literature only, to make sure of high scientific quality standards in the selected articles (Räsänen et al., 2016). In those articles, a scientific methodology is employed to test empirically research hypotheses and to be sure that data gathering, treatment and results analysis are logical, verifiable and reproducible. Finally, only English-written literature was included in order to keep references that can be widely read, to enable reproduction, and to deepen this review. The search engines used for the literature review process are ISTEX, Science Direct, Scopus and Web of Science, four traditional academic citation databases. In addition, we completed the analysis with the literature extracted from Google Scholar database. Despite it has been shown that the majority of literature identified using Web of Science was also found using Google Scholar (Haddaway et al.

2015), it has been also demonstrated that a large fraction (9–30%) of highly-cited documents in the Social Sciences and Humanities could be invisible to Web of Science and Scopus (Martín-Martín et al. 2018).

The initial search used inclusion criteria to be sure the first selected range of articles focused on studies about vulnerability to cyclonic floods in Bangladesh. The following search sequence used in the electronic databases includes the hazard, the area of study and key words related to vulnerability: *("coastal flood\*" OR "sea-level*

*rise" OR "Storm surge" OR "Cyclonic storm" OR "Disaster risk reduction") AND ("Bangladesh" OR "Brahmaputra delta") AND ("\*vulnerability" OR "human exposure" OR "coping\*").* Those terms have been sought in the titles, abstracts and keywords of the articles (see Appendix A1 for more details on the strategy used). A second selection of the first range of articles screened documents one by one to check if they explored vulnerability to cyclones and if they specifically did focus on areas placed in coastal Bangladesh, and not

vulnerability to river flood for instance. From this research, 49 publications were kept for this review (see Appendix A2- A4).

## 4.2 Summary of the literature analysis

The first result shows the universities behind the analyzed articles: 85% of the studies were conducted by Western or Asian scholars. Only seven articles were directed by Bangladeshi nationals, and the rest is made from a

250 cooperation with Bangladeshi. Amongst the authors' disciplines, the applied sciences are overrepresented with 26 articles (especially in the field of disaster risk management), followed by the humanities and social sciences and, finally, the natural sciences.

The methodologies used in half of the studies are mixed methods, using both quantitative (e.g., surveys) and

255 qualitative data (focus group discussion and key informant interviews, for example). Inductive approach (bottom-up) was used in three quarters of the articles, explaining that most of the studies were done at a small scale - villages and *unions* (a gathering of two or three municipalities) – to study some aspects of population vulnerability. While still more than half of the studies used surveys (structured questionnaires), five articles used the Participatory Rural Appraisal (PRA) method to allow the participant to fully contribute even if they were illiterate

(Ahmed et al., 2010; Ahsan, 2017; Shameem et al., 2014).

Most of the authors touch upon vulnerability in their articles, but surprisingly, not all give a definition of it, often taking the concept for granted. Among the studies defining the theory upon which they grounded their research, three articles define what they consider as being vulnerability. First, for Mallick et al. (2009), "*vulnerability to*

265 *disasters refers to the inability of a society and its people to withstand adverse impacts from multiple stresses to which they are exposed. It has two sides: an external side of risks, shocks, and stress to which an individual or household is subject; and an internal side which is defenselessness, meaning a lack of means to cope*". Second, Quader et al. (2017) mention "*three main characteristic groups of social vulnerabilities (dimensions), namely the socio-demographic and economic characteristics of the population, their access to basic facilities and the*

270 *proportion of physically or ethnically marginalized people*". Then, Ishtiaque et al., (2019) define vulnerability "*as the degree or extent to which a system is likely to be exposed and sensitive to a hazard, and the capacity of that system to adapt to the effects of climate impacts*". Although those definitions do not have a lot in common, they are complementary. Therefore, those results show how little consensus exists on this concept in this region.

Finally, this literature review enables us to outline below the vulnerability factors' main characteristics discussed in the articles (physical and systemic infrastructures, social and economic aspects). Some authors agree that in the coastal zone globally, but especially in Bangladesh, it is imperative to consider both physical and socio-economic factors in defining vulnerability (Miah et al., 2020; Mullick et al., 2019; Uddin et al., 2019a).

### 4.3 Identifying vulnerability indicators

**4.3.1 Physical and systemic infrastructures**

**4.3.1.1 Infrastructures: Road, cyclone shelter, embankment and communications facilities**

According to the authors, the lack of proper **road** communication networks and transport infrastructure is an important factor of people's vulnerability to cyclone disasters. In the coastal area, roads are not uniformly distributed (Rabby et al., 2019), most of them are narrow, unpaved and made of dirt tracks, since they often mark

boundaries for crop fields (Alam and Collins, 2010), and embankments are also used as roads. During the rainy season, they are destroyed and the mud makes any physical mobility difficult (Saroar and Routray, 2010). The lack of road infrastructure is particularly important when authorities should disseminate the early warnings before a cyclone (Paul, 2009), but they are mainly inaccessible to motorized vehicles. When the hazards are imminent, roads in good shape are needed for the population to travel to reach the protective shelters (Kulatunga et al., 2014).

Finally, after the cyclone, developed road communication networks enable effective evacuation and rescue during post-disaster relief operations (Hossain, 2015). Hence, it is concluded by authors that remote areas rarely get emergency assistance (Mallick et al., 2017).

Situation of the **shelters** are studied by many authors, showing how the long distances to reach them, the lack of

infrastructure facilities, and their deterioration is an issue to the nearby households when they have to decide to find shelter or not. Proximity, poor hygienic condition (Kulatunga et al., 2014), lack in communication means and delivery of sanitation and drinking water (Saha, 2015) are some example for additional reluctant aspects for many of the families. People are also reluctant to take refuge in shelters because they are suspicious of cyclone alerts, but also because they are afraid of having all their belongings stolen. There is also a memory loss of the past events

impacts (Ishtiaque et al., 2019). If each shelter can accommodate approximately 1000 people (Mallick and Vogt, 2011), they are often insufficient to host the dense population living in the area, not uniformly distributed on the territory or are also too far from the houses in need (Alam et al., 2020; Hossain, 2015). The distance to a shelter may considerably increase the vulnerability of the people if they decide to walk to one (Alam and Collins, 2010).

The **embankments** are often in very bad maintenance, including the sluice gates (Younus and Sharna, 2014). Embankment failure can also induce problem with drainage congestion and water logging (Mullick et al., 2019). Those coastal embankments are also breached on purpose to be penetrated by pipes that fill the basins of shrimp farms (Hossain, 2015).

It is worth noting that in the small island polders there is no **electricity** and, in remote areas, it is harder to find **hospitals** or clinics as well as available doctors, because they are located mainly in big towns and cities (Garai, 2017; Islam et al., 2014b). Those aspects definitely decrease people's coping capacities.

### 4.3.1.2 Houses: Material and emplacement

**Houses' materials** and emplacement are the main criteria of the household's vulnerability. Four main kinds of dwelling houses may be described in coastal Bangladesh. The *pucca* houses are the most solid and permanent houses, made of brick and concrete, they are very rare and held by wealthy households. The *semi-pucca* buildings are made of semi-permanent material in the sense that they have a floor of mud, brick or cement, but their walls are bamboo mats or timber and the roof of corrugated iron (CI) sheets. The *kutcha* houses have floors made of mud; the walls are bamboo, sticks or straw; and the roof made of paddy (rice) or wheat straw (Islam and Walkerden, 2015). Finally, the *jhupri* houses are mainly huts or made of CI sheets. *Semi-pucca* houses are not cyclone resilient, but *kutcha* and *jhupri* houses are organic-made structures totally vulnerable to this hazard (Miah et al., 2020; Parvin and Shaw, 2013). In the majority of the rural studies analyzed in the literature review, most of the houses are kutcha houses (Ahsan and Warner, 2014; Akter and Mallick, 2013; Younus and Sharna, 2014). Houses' material fragility is accentuated by their settlement place: the more the villages are isolated and scattered, the more they are vulnerable.

Moreover, there is a dreadful situation when houses are settled along the embankments: they face huge casualties and damages with strong winds and waves, or in case of any breach of dykes (Hossain, 2015). This also happens for people living out of the dykes: when their living places remain flooded in the days and weeks following a cyclonic inundation, people take shelter on embankments, building tent-like huts (Garai, 2017; Mallick et al., 2011). The main reason explaining people's choice to install on embankments is because they are the major public spaces owned by the government along the coast which have become the last area above sea level not privately owned where displaced people may construct settlements (Alam and Collins, 2010).

### 4.3.1.3 Water: Salinity, sanitation and drinking water

Saline intrusion due to cyclonic inundations is a serious problem highlighted by the coastal population, both for food and sanitation. Because of the lack of embankments' reconstruction after cyclones such as Aila in 2009, saline water intrusion was still a problem one year later (Mallick et al., 2011). **Salinity** is especially dreadful for arable land, where farmers have to wait many years before restarting rice cultivation (Ahsan and Warner, 2014; Ishtiaque et al., 2019; Mallick et al., 2017; Rabby et al., 2019).

Many authors highlight that after cyclonic inundations, **sanitation** facilities are particularly challenged. The first main problem is the drinking water supply, which is disrupted and becomes scarce (Alam et al., 2017; Younus and Sharna, 2014). Polluted by waste or animal carcasses (Hossain, 2015), as well as human feces due to the lack of sanitation (Parvin and Shaw, 2013), the water is often contaminated by water-borne diseases like diarrhea, or even cholera and typhoid (Ahsan and Warner, 2014).

**Drinking water** is also contaminated by saltwater intrusion, which is a serious problem for most the families that use pond springs (Das et al., 2020; Ishtiaque et al., 2019; Shameem et al., 2014). Besides failures in access to sanitation and clean water, households are significantly disrupted, facing even long times with no electricity after a cyclone (Akter and Mallick, 2013), which can be an issue for communication purposes and economic recovery. It is worth noting that in the small island polders there is no electricity at all.

### 4.3.2 Social and cultural factors

A significant indicator influencing people's vulnerability is **education**. It was shown that less educated people were less able to understand the disaster forecast and to foresee the appropriate materials and food to store in case of evacuation (Paul and Routray, 2011). The analysis of the literature review establishes that five main social groups are particularly vulnerable during cyclonic inundations: **women**, **children**, **elderly**, **disabled people** and **religious minorities** (Das et al., 2020; Mullick et al., 2019; Rabby et al., 2019). Firstly, in customary social order in Bangladeshi rural areas, men are usually the heads of household, being the ones who decide how to prepare the family for a cyclone, where to take shelter and if to eventually evacuate from the house (Ahmed et al., 2010). Traditionally, women take care of family resources and have to wait for their husbands the consent to go to a protective shelter (Alam and Collins, 2010; Kulatunga et al., 2014). During the cyclone and post-disaster, large families (more than 10 members) are considered as more vulnerable, in the sense that elderly, children and people with disabilities are considered to be dependent people, for the evacuation as well as in the recovering phase (Hossain, 2015; Quader et al., 2017). Generally, women have less access to incomes and assets because social norms limit their advocacy in the public sphere, although they can get access to microcredits offered by NGOs to support their recovery. Nonetheless, authors pointed out that financial relief is often distributed to particular religious, political and social groups, neglecting women, poor and religious minorities, a situation that mainly gives advantage to the supreme social class (Mallick et al., 2017). This social class, in rural zone, corresponds to the wealthy persons who regularly become elected representatives in the local government (Mallick et al., 2011). However, some NGOs emphasize that policies aimed at improving infrastructure to reduce vulnerability must be accompanied by a policy of sensitizing populations to the disaster in order to reduce illiteracy, poverty and increase livelihoods (Ishtiaque et al., 2019).

Beyond these social indicators, other cultural and/or political variables are mentioned in the literature. For example, despite the Cyclone Preparedness Program, many critics were found in the literature about the early warning system's effectiveness, either because of its dissemination or its understanding. The first problem in the preparedness phase is the population's access to the alert, since a large part does not have access to a **radio**, the communication dissemination remains poor (Ahsan and Warner, 2014; Mallick et al., 2011; Paul and Routray, 2011; Saha, 2015). Moreover, several researchers noticed the surprise of the population at the arrival of the cyclone disasters because they failed to understand the warning systems (Mallick et al., 2017; Younus and Sharna, 2014). Other authors also report people are indifferent to the weather forecast because they use their own traditional forecast methods through indigenous signs (Hossain, 2015; Rakib et al., 2019). Therefore, signs in the environment (wind directions, clouds, sea birds and insects' behaviors) are used instead to perceive impending hazards (Garai, 2017). When the information of the early warning has been effectively transmitted to the people and they are aware of the imminent hazard, many scientists underline the population's resignation that a cyclone is God's will. They regard disasters as "common events" (Kulatunga et al., 2014) since they happen frequently. In case of total flooding and destruction, people may end up having to move to another place. Nonetheless, authors observed that if people can temporarily move out of their village, they do not consider permanent relocation as alternatives (Ahmed et al., 2016). Some authors argued that the polders "create a false sense of security" (Sultana, 2010) and therefore increases people's vulnerability (Ishtiaque et al., 2017). This feeling of security can also diminish the efficiency of a cyclone warning system. In addition, authors note that memory loss from previous extreme events can be observed (Ishtiaque et al., 2019).

### 4.3.3 Economic factors

Some labor categories, tied to the coastal economic development, are more vulnerable to cyclones, namely rice farmers, fishermen, and shrimp farmers. Cyclones particularly affect cultivated lands and fishing facilities, often leading to extreme poverty for the people who cannot practice their job anymore (Mallick et al., 2017). Shrimp farmers are especially concerned with this problem, where a quarter of them suffered more damages than agricultural farmers after cyclone Aila in 2019 (Ishtiaque et al., 2017). The expansion of shrimp farms in Bangladesh, especially in the West and East coasts have raised a certain number of issues both for the people and for the environment. This culture has been developed mainly in the three last decades because of its great potential of exportation, bringing back value from foreign currency, until it became one of the most important sectors in the country especially concentrated in the coastal zones of Khulna, Satkhira, Bagherhat and Cox's Bazar (Ahamed et al., 2012).

Finally, to conclude this diagnosis, let us stress that according to Uddin et al. (2019a), the variables of population density, poverty and geomorphology (coastal and flat/low-lying country face to sea level rise) will be the most aggravating factors of future vulnerability. This literature review allows identifying the main characteristics to assess the vulnerability. While from a theoretical point of view, it can be defined on the basis of all these physical, socio-cultural and economic indicators, its spatial component is still not addressed and the interrelation between a population and its territory of life is not analyzed or taken into account here. Yet, the spatial distribution of its different components (physical, socio-cultural and economic) enables defining its level for a territory. This is why the index approach was chosen because it allows synthesizing all this information into a single variable and mapping it.

### 5 A socio-spatial vulnerability index

To identify places that are highly vulnerable to cyclonic flooding hazard, we focus on 16 coastal districts, the district being the spatial unit of reference (Fig. 1a). As widely discussed by the Program Development Office for Integrated Coastal Zone Management Plan (PDO-ICZMP, Uddin & Kaudstaal, 2003), the definition of a coastal zone is not simple and depends on selected criteria. According to the PDO-ICZMP classification, we have considered 12 "exposed" districts, i.e. ones adjacent to the sea and/or located in the lower estuaries (i.e Khulna, Satkhira, Barguna, Cox's Bazar, Bagerhat , Patuakhali, Pirojpur, Chittagong, Noakhali, Bhola, Lakshmipur and Feni). The interior coast districts (Jessore, Narail and Gopalganj) are less exposed to cyclone inundations, as the other districts facing the sea. Moreover, following the literature review (Fig. 1a, see Appendix B.b) and given the trajectories of major cyclones as Bhola, Gorky or Sidr, we considered that the districts in the mouth area, i.e Barisal, Shariatpur and Chandpur, are also highly exposed to cyclonic floodings. Finally, the district of Jhalokati, cited in the literature review and bounded by Barisal, Pirojpur and Barguna, is added to maintain a territorial coherence.

There are two popular approaches to constructing a vulnerability index: the variable reduction and the variable addition. The first one is an inductive approach, a large set of variables is used, assuming that they have potentially larger or lesser influence (i.e. weight) on the index calculation. Principal Components Analysis, although fairly complex, is often used to estimate these weights to be assigned to each indicator (Cutter et al., 2003; Das et al., 2020; Quader et al., 2017; Uddin et al., 2019a). Although a large number of variables may be useful for descriptive

purposes, including non-influential variables in the index, aggregation may decrease both explanatory power and easiness of its use and understanding. On the contrary, the variable addition is a deductive approach. Deductive models can contain a few dozen variables, or less, which are normalized and aggregated to index, which could be separated into groups sharing the same underlying vulnerability dimension. This approach is the most common structure applied to vulnerability indices. Deductive approach, based on a data-driven mindset from expert knowledge (e.g literature review) and parsimony, helps to identify and trace underlying themes running through the data, opposite to inductive approach where the statistical models obfuscate underlying data. The literature review is a rich source to understand the main causes, translated as indicators of vulnerability, as well as their relative importance and interactions. Therefore, we chose to apply this simple method, based on the vulnerability factors deduced from our literature review, to determine the SSVI.

### 5.1 Methodology

From the literature review (section 4) three components based on local experts' knowledge are highlighted: physical and infrastructures, socio-cultural and economic components. Like explained above, exposure is included in the vulnerability, a fourth component is added: cyclone protection and exposure.

The calculation of SSVI is based on the methodology developed by Flanagan et al. (2011). In order to construct it, each of the variables selected was ranked from highest to lowest vulnerability degree. A percentile rank was then calculated, by using the formula Percentile Rank = (Rank-1)/(N-1) where N is the total number of data points, for each district over each of these factors. In addition, a percentile rank is calculated for each of the four components or domains (adapted from Flanagan et al. (2011)), defined as (1) socioeconomic status, (2) household composition and disability, (3) Housing & Infrastructures and (4) Cyclone Protections and Exposure, based on a sum of the percentile ranks of the factors comprising that domain (see Appendix C for full methodology). Finally, an overall percentile rank for each district is calculated as the sum of the domain percentile rankings and defined as the SSVI. The interpretation of the SSVI is as follows: for example, a district being in the 85th percentile (0.85 ranking), it means that 85% of these districts are either below, or equal, to that particular district regarding to SSVI.

### 5.2 Data and sources

From the literature review, 17 variables were identified to assess the level of vulnerability of each coastal district (Fig. 2). The representativeness of each variable, defining the four components, is characterized according to the frequency they were cited in the literature review. The frequencies are in the range of 12% (Minorities) to 47% (Education), and the median is ~27% (Children), i.e. 13 studies about the 49 considered (for more details see Appendix B).

All the data considered here are freely available, accessible online and given at district scale. These data come from different national sources such as the Bangladesh Bureau of Statistics (2011 and 2014 surveys) and from international sources such as the World Bank. Full details of the data used are in Appendix D.

To describe the **Socio-economic status**, two variables are used: *poverty rate* which corresponds to percentage of total of poor comprising both lower and upper the poverty line; and *education rate* which corresponds to

percentage of children who have more than 10 years of school (Alam et al., 2017; Islam et al., 2014a; Ishtiaque et al., 2019).

The **Household composition and disability** domain is characterized by 5 variables (Alam et al., 2020; Ishtiaque et al., 2019; Rabby et al., 2019): *Age 14 or younger* (%) and *age 60 or older* (% ) (ages 14 and 60 are chosen as limits according to Akter and Mallick (2013)) correspond to addition of categories between ages 0-14 years and categories above 60 years, respectively and expressed in percentage, *female rate* (%), *disability* represents the percentage of disable people per district and *religion*, described by percentage of Muslim and percentage of religious minorities included Buddhist, Hindu, Christian and other (Akter and Mallick, 2013; Das et al., 2020; Garai, 2017; Ishtiaque et al., 2019; Rabby et al., 2019; Roy and Blaschke, 2015).

The third domain, called **Housing and infrastructures**, includes percentage of *houses made of organic materials* (*kutcha* and *jhupri* that are structures totally vulnerable to coastal flooding (Parvin and Shaw, 2013)), *unpaved road* corresponds to percentage of length of road that are not paved, *access to electricity, mobile and sanitary* (toilet with water and sealed) represent the percentage of population which have access to these infrastructures, *unsafe drinking water* represents percentage of population which have not access to safe drinking water (unsafe drinking water corresponds to all spring water that are not coming from the tap or from tube wells) and then *number of hospitals* per district (Ahsan and Warner, 2014; Das et al., 2020; Ishtiaque et al., 2019; Islam et al., 2014a).

The last domain **Cyclone protection and exposure** is described by three variables: *dykes and embankments* (Alam et al., 2010; Hossain, 2015; Rabby et al., 2019) which corresponds to the length of dykes and embankments into each district, *shelter capacity* computed by the difference between the shelter capacity (number of people) and the number of people present within a one-kilometer radius of the shelter; and then, *exposure* corresponding to percentage of people per district concerned by the cyclonic inundation with a 50-year return-period (Fig. 1). As we have seen above (section 3), this cyclonic inundation probability, resulting from hydrodynamical modeling, integrates components as topography, bathymetry, slope, dikes, embankments, soil roughness and vegetation, which allows to indirectly integrate the physical vulnerability in the SSVI. Although authors proposed population density as a variable to characterize the vulnerability (Ahsan and Warner, 2014; Ishtiaque et al., 2019; Kulatunga et al., 2014; Miah et al., 2020), none used the population density actually affected by cyclonic floods.

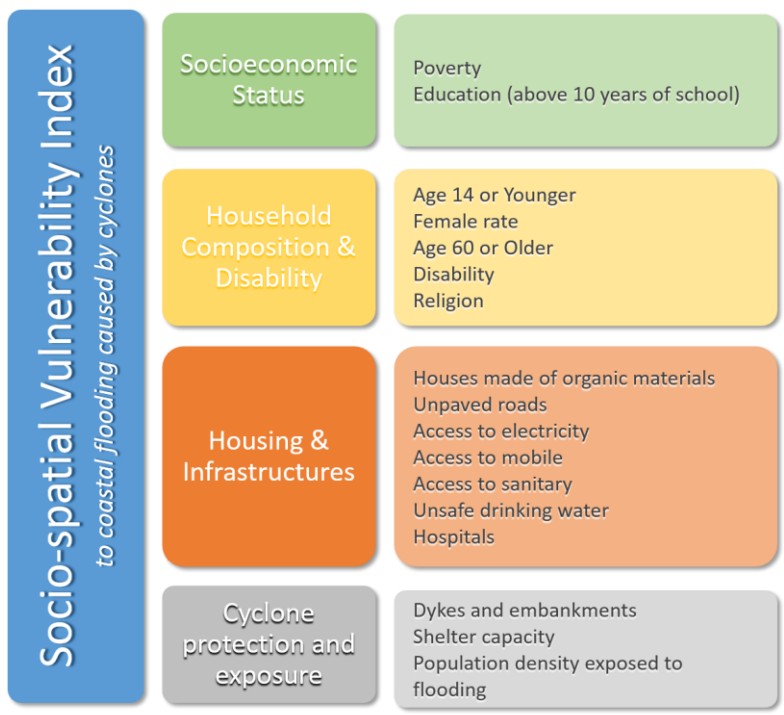

**Figure 2: Socio-spatial vulnerability index scheme (adapted from Flanagan et al. 2011)**

**5.3 Socio-spatial vulnerability index to cyclonic flooding mapping**

Figure 3 presents the SSVI. We deduced that the inhabitants of the districts at the mouth of the GBM rivers, i.e. Chandpur, Shariatpur and Barisal, and Bagerhat in the west coast, where the SSVI is in the highest category, are more vulnerable to cyclonic flooding relative to other districts. It appears clearly that these districts are very densely populated, poor, with a low percentage of high school completion and extremely exposed to the inundation hazard with insufficient cyclone protections. However, this analyze does not mean that the population of other districts are not vulnerable, but only less vulnerable.

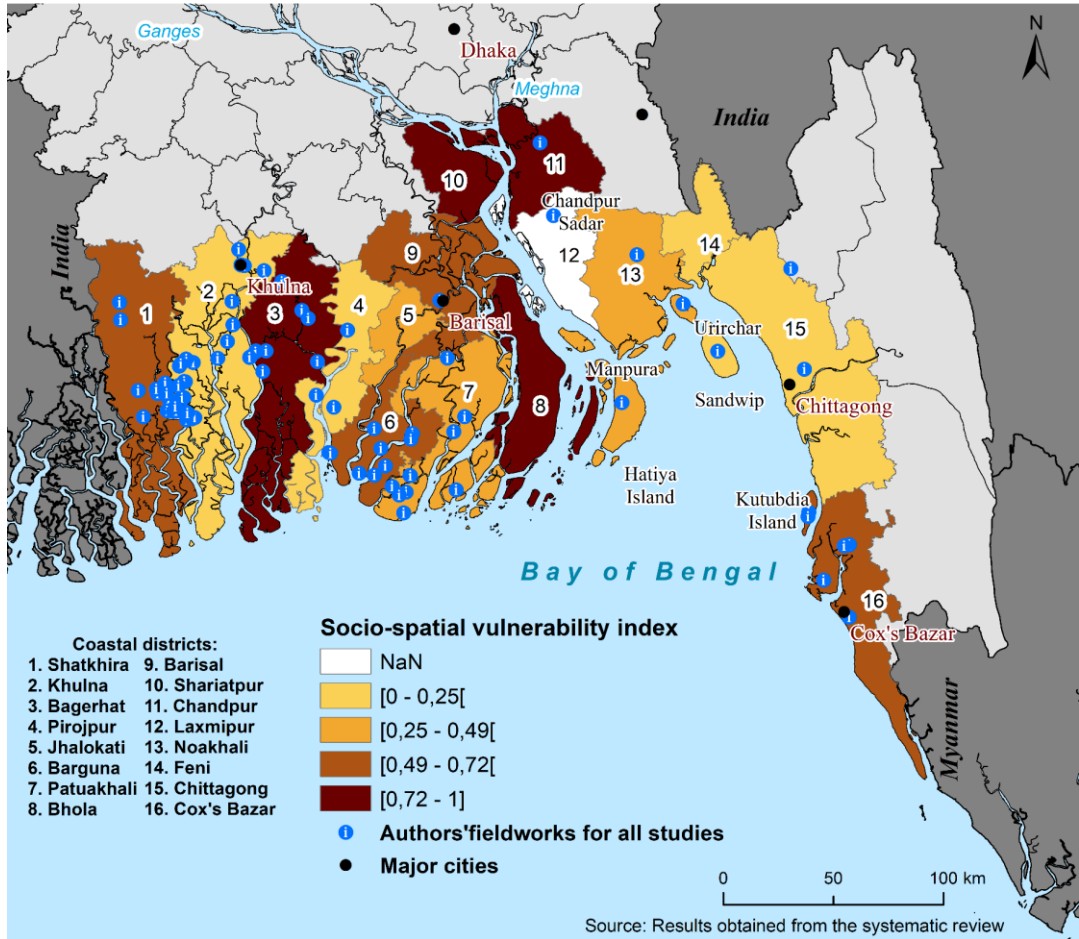

**Figure 3: Socio-spatial vulnerability index to cyclonic flooding and location of case studies upon which are based the articles included in the literature review.**

For the districts that appear the least vulnerable according to the index (Fig. 3), the *Cyclone protection and exposure* domain is the one that appears the most critical for these districts, as for example, Feni, Khulna, Pirojpur and Patuakhali (Fig. 4). Then, it would appear that the *Household composition and disability* domain is the most vulnerable domain for the district where the vulnerability level is moderate (Borgona, Shatkira). Concerning the last two most vulnerable districts (Chandpur and Shariatpur), the three domains *Socio-economic*, *Household composition and disability* and *Cyclone protection and exposure* have very high values, exceeding 0.8.

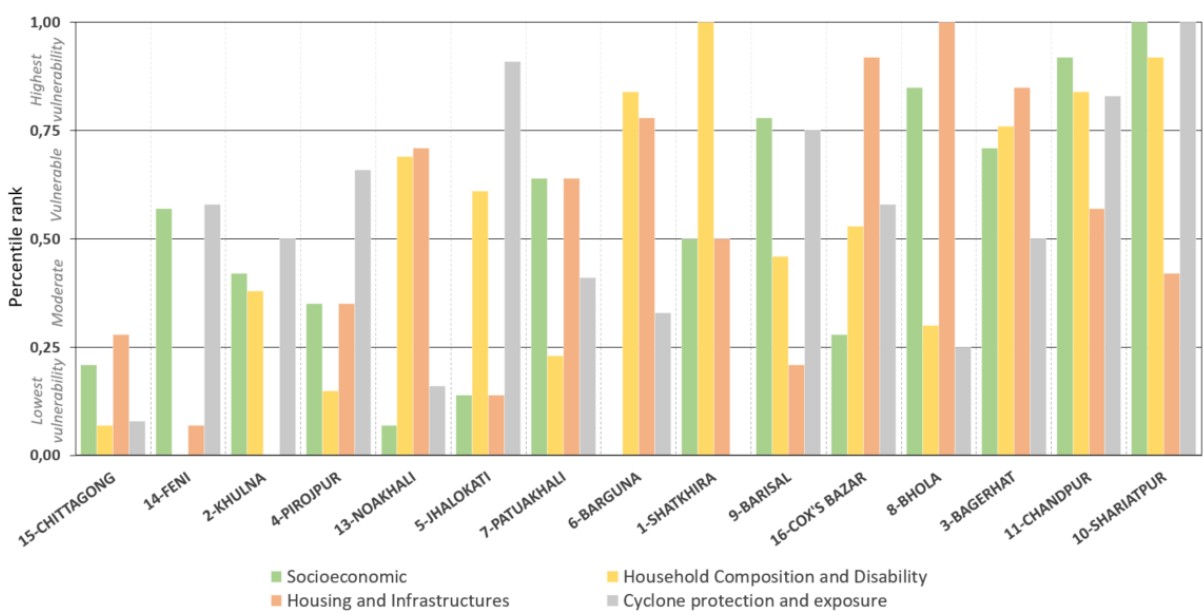

**Figure 4: contribution of each domain to the socio-spatial vulnerability level.**

## 6 Discussion

### 6.1 Main findings and contributions

Although considerable effort has been made in Bangladesh, as shown in the literature review (section 4), to understand patterns of social, economic and environmental variables that render inhabitants more or less vulnerable to cyclonic flood, much less effort has been spent to consider the socio-spatial vulnerability. This study provides, from an interdisciplinary approach, a methodology and an integrated index, called 'socio-spatial vulnerability index' (SSVI), for assessing the ways in which socio-spatial vulnerability to cyclonic flooding is distributed across the coastal territory. This new metric is a function of the probability of the cyclone flood hazard and the sensitivity of inhabitants.

Compared to recent studies on vulnerability assessment, our approach differs in several aspects. First, our study is intended to be general in scope, considering the entire population and territories affected by the cyclonic flooding, in contrast to studies that target either a category of population (Alam et al., 2020; Swapan et al., 2020), or a particular site, at the village scale, for example in Rakib et al. (2019). These highly localized studies have the advantage of being more detailed and enriched with qualitative observations on the perception and representation of natural hazards, although not generalizable to the scale of coastal districts. These studies are useful at the local level for strategic decision-making and orientation of risk management but remain too specific for the implementation of such strategies at the national level like the deployment of the Delta Plan 2100 (Ministry of Planning, Government of the People's Republic of Bangladesh, 2018).

Our study offers an analysis of the vulnerability to a hazard, the cyclonic flooding. It does not voluntarily take into account information on the adaptive capacities of populations and territories as proposed by Uddin et al. (2019a). Adaptive capacity, as well as resilience, are fields of study in their own right that must, in our opinion, be distinguished from vulnerability. Indeed, the latter authors integrate, among other things, the presence of local and private banks and the possibility for small farmers to make loans, enabling them to restart their activity after a natural disaster. This information impels to define the capacity to cope of these populations rather than their vulnerability, as distinguished by Quader et al. (2017).

On the other hand, we argue it is essential to integrate the exposure of populations and territories to a hazard into the definition of vulnerability. A population, whatever its socio-economic and demographic characteristics, is not vulnerable if it is not exposed. Contrary to Rabby et al. (2019) and Das et al. (2020) who assess vulnerability solely on the basis of social factors and infrastructural factors (quality of housing, access to drinking water, electricity, sanitation, and so on), the SSVI that we provide integrates the exposure to the hazard. The population densities actually affected by the floods as well as their possibility to find shelter or to be protected by defense structures are essential and indispensable information to the evaluation of the socio-spatial vulnerability. In many studies, only the whole density of population is used to represent exposure (i.e. Ishtiaque et al, 2019; Das et al., 2020), without distinguishing the population actually affected by cyclonic flooding.

One of our goals in this study is to produce transferable information for decision makers. This is why we chose an approach that combines all the dimensions of vulnerability. We believe the territorial approach is essential in the decision-making process. The first step is to identify which regions are the most vulnerable to a hazard before identifying which dimensions of vulnerability need to be improved. Our results could help out the deployment of the Flood risk Management Strategies of the Delta Plan 2100 on the districts identified as failing on this dimension of vulnerability. Sub-strategies FR 1.1, 1.2 and 1.3[1] for example, correspond to the Cyclone protection and exposure dimension of the SSVI, for which Shariatpur and Jhalokati districts appear to be the most vulnerable (see figure 4).

The approaches proposed by Uddin et al. (2019a) for instance, present a mapping of vulnerability by dimensions: demographic vulnerability, economic vulnerability, agricultural vulnerability, and so on. Deprived of a synthetic map, it is not clear if the decision-maker can decide where and how to intervene on the territory. Other studies are specific to a single dimension of vulnerability, as physical vulnerability (Islam et al., 2016; Islam et al., 2015; Hoque et al. 2019) or social vulnerability (Rabby 2019, Das 2020). Nevertheless, the SSVI empowers to integrate all the dimensions of vulnerability and provides usable information for decision makers.

Comparing vulnerability studies of coastal districts exposed to cyclonic flooding risk remains very difficult because the definition of the vulnerability concept varies greatly from one study to another. Quader et al. (2017) is certainly the recent study reporting the most relevant definition and assessment of vulnerability to our study. The results corroborate that the districts in the mouth of the Meghna (central coast) up to Chandpur and Shariatpur districts, as well as the Bagerhat and Cox's Bazar districts are highly vulnerable. The only notable difference is situated in Shatkhira district, which is vulnerable according to our study, while it is very low vulnerable according to Quader et al. (2017). In detail, the different dimensions of vulnerability are not described in the same way by the two studies. For example, accessibility to electricity is used to define demographic and basic facilities vulnerability in Quader et al. study, while it is used to define the vulnerability of infrastructure and housing in our SSVI. Likewise, disability is either used to define household vulnerability (SSVI), or considered as a separate dimension of vulnerability (Quader et al., 2017). Moreover, we used a robust probabilistic cyclonic flood hazard map based on a dataset of 3600 statistically and physically consistent synthetic cyclone events (Emanuel et al., 2006; Khan et al. 2019). While Quader et al. (2017) used a low level of confidence cyclone hazard density

---

[1] Protection by development and improvements of embankments, barriers and water control structures (including ring dikes) for economic priority zones and major urban centres; construct adaptive and flood-storm-surge resilient building; adopt spatial planning and flood hazard zoning based on intensity of flood.

interpolated map based on historical cyclone tracks (~160 events in 1877-2015). Although the level of detail provided by this union-wide study is significant, the closeness of the results is meaningful. However, social vulnerability is defined from 141 variables and a consistent workflow with several statistical methods. On the contrary, in our study we dedicated a strong consideration to the theoretical links between indicators and underlying vulnerability to cyclonic flooding by conducting a strict literature review. Therefore, SSVI is computed from only 17 variables and a simple computation of index appears easier to understand its construction and to replicate by decision makers.

**6.2 Representativeness and quality of the data at the district scale**

A first limitation concerns the availability of dataset to construct the SSVI. The dataset used for this study was produced at different dates: from 2010 for the poverty level, to 2018 for paved/unpaved roads. However, the main information defining social vulnerability comes from the BBS Census of 2014, which gives some homogeneity to the description of the socio-demographic characteristics of the population.

Besides, a few variables used to calculate the SSVI have some underlying assumptions that might differ from reality and may give a false representation of the situation on the field. For example, we assumed that dikes and embankments protect people and agricultural production, but everything depends on their condition, maintenance and breach presence (Younus and Sharna, 2014; Mullick et al., 2019; Hossain, 2015). In recent studies, dike breaching is found to be the reason behind flooding (e.g., Hossain 2015, Adnan et al. 2019, Khan et al., 2020). For example, Hossain (2015) mentioned that dikes and embankments may be in very poor condition and may not perform its protective function. Similar results are suggested by Khan et al. (2020) from analysis of the recent cyclone Amphan that made landfall in May 2020. Additionally, based on the results presented in Adnan et al. (2019), one might argue that the dikes were not adequate to protect against a certain event (either because of their design, or by gradual degradation of the dikes). As noted in this article for Sidr, if the dike has not had breached, the inundation would have been 18% of the coastal area, whereas the dike breach increased the number to 35%. Unfortunately, current dikes condition information is not available nor accessible in public data, and according to our understanding, not monitored either. The assumption in our hydrodynamical modelling is that the dikes are in well serviced condition, e.g., dike breaching (which is a different geotechnical process, and far from the scope of this paper) was not modelled. Furthermore, after a cyclonepopulations can and do settle on dikes, sometimes the only free public spaces left to the people, making them very exposed (Alam and Collins, 2010). Undoubtedly this issue about the balance between the negative and the positive contribution of dikes and embankments should be taken into account in future studies on vulnerability to cyclonic flooding in this region.

The shelter capacity is another example of variable that does not appear always representative of the situation on the field. It can be assumed the presence of cyclone shelters reduces people's exposure and therefore their vulnerability. However, the presence of cyclone shelters in close proximity to homes, within a radius of 1 to 1.5km, does not mean that they are useful and used. Like mentioned by Mallick et al. (2017), shelters are not optimally placed on territories in order to be easily accessible and accommodate as many people as possible, but are rather situated near the supreme classes dwellings. These buildings are not always maintained and do not meet the requirements of the local society: men and women are mixed, there are no women-only sanitary facilities, and people may feel in insecurity (Kulatunga et al., 2014; Saha, 2015). Accessibility to cyclone shelters is also conditioned by the state of the roads to get there. Therefore, we chose to represent the road condition by the variable "paved or unpaved road". However, we can consider this information is valid for estimating vulnerability

only before the passage of a cyclone (in the prior-disaster phase) because the roads can be very damaged afterwards, as mentioned by Saroar and Routray (2010), as well as during the rainy season (impacting even more the post-disaster phase).

The sub-district level case study presents advantages in data mining, the use of participatory methods and in capturing the fine complexity of vulnerability, but these studies are very local specific and therefore are difficult to compare with other regions. Moreover, the integration into a vulnerability index of different local qualitative and quantitative data, which have their own logic of underlying scales and pattern, implies a loss of information and relevance (Fekete et al. 2010).

Here, using the "district" scale is related to the multifold aims of our research. One of the objectives was a spatial identification of hotspots of vulnerability to cyclonic flooding, intending both to identify the spatial variability linked to cyclonic flooding facing the district and the districts where the combined effects of multiple social, economic and environmental stressors are most prevalent regarding the cyclonic flooding. Therefore, the districts have been chosen as units of analysis for several motives: 1) districts are relatively homogeneous in size in comparison with sub-districts, municipalities or unions; 2) cyclonic hazard management are organized and supervised on the district level (e.g the alert and cyclone warnings dissemination system, maintaining cyclone shelters and assisting in evacuation procedures, Kulatunga et al., 2014); 3) a sufficient number of variables is freely available online from the Bangladesh Bureau of Statistics, 4) district unit can be more easily transposable and replicated on other delta regions (e.g applied in the 13 provinces of the Vietnamese Mekong Delta or in the 6 districts of the Ayeyarwady Delta); and 5) as administrative unit, the districts are easily understood by decision makers, planners and end-users.

Moreover, using this territorial delimitation enables us to zoom out the research fieldworks conducted from 2007, in order to show which zones have been under the scope of science and others very exposed and vulnerable but almost forgotten by researches. While most of the analyzed studies worked on local places, they did not give a whole picture to the reader about the work made in coastal Bangladesh. The distribution of the studies on the map (Fig.3) shows that two main districts on the west coast were over-studied (Shatkhira and Khulna) compared to the need of studying now the most vulnerable districts, situated in the mouth of the Delta.

**6.3 Limits of the research**

A significant limitation to this work is the approach intrinsically used in this literature review: the exclusion criteria left out the analysis of grey literature to focus solely on peer-reviewed articles. By doing so, we excluded reports from local NGOs in Bangladesh that could bring to light new elements only known at the local scale after a long period of time. Nonetheless, we obtained a clear view of what is done throughout scientific articles in terms of coastal vulnerability. Additionally, we obtained 42 out of 49 analyzed articles which involved either Western or Asian scholars likely to assign concepts to Bangladesh that would normally work only into their own foreign country. Thus, the outcome of socio-spatial vulnerability is defined mainly from a Western perspective, because the majority of the articles included Western researchers.

Moreover, it should be noted the results of the top-down approach used in this paper were based on aggregated data from studies that employed different methodologies. Consequently, a bias could occur since by aggregating the data, we lost some accurate information and perhaps considered a concept defined with the same word but conceptualized differently between different studies. Another issue raised with the top-down approach is the

absence of local community participation in this assessment. This feature is necessary to give a complete validity to such an evaluation, by taking into account people's opinions and perceptions about their own vulnerability (Rakib et al., 2019; Sattar and Cheung, 2019). It is also through their participation that possible solutions and strategies of adaptation to disasters may be formulated (Pouliotte et al., 2009).

However, in this study, the objective of the SSVI mapping exercise is to open a dialogue around vulnerability to cyclonic flooding in Bangladesh, and to help stakeholders and researchers to identify the most vulnerable places that are understudied. Therefore, as the SSVI objective is not to participate at adaptation practice or decision-making, the validation question, while important, is not central (De Sherbinin et al. 2019). Moreover, as the SSVI is multidimensional and not directly observable, and because there are few published explicit procedures that outline how to validate it, this poses a persistent challenging validation question (Tate 2012; Rufat et al. 2019). Moreover, the human and economic cost from the cyclone impact reports, if available, are unusable to the SSVI validation because they address mainly the exposure component and undervalue the vulnerability. Nevertheless, some vulnerability index validation research tracks seem to emerge from them: 1) external validation from independent proxy data as death tolls, physical wounds, diseases, economic loss, and household survey; 2) internal validation from sensitivity analysis. Future SSVI to cyclonic flooding in Bangladesh research should investigate validation scheme based, for example, on qualitative work describing social stratification in pre- and post-disaster settings (Fekete et al. 2019) and from targeted surveys, and participatory approaches which would be conducted in different districts.

Additionally, we pointed out that vulnerability for people nature-dependent to the Sundarbans Forest was underestimated. Moreover, in this prawn farming region, it is amplified by the illegal or forced occupation of land that often lead to violence. Vulnerability is also stressed by intensive, potentially hazardous and poorly paid working conditions, especially for women (Ito, 2002; Paul and Vogl, 2011). Nonetheless, specific populations as the ones living in slums or Rohingya refugees are under-represented in our study (Alam et al., 2020; Swapan et al., 2020). In the same way, it should be noted that there are a lot of new proxies of social, environmental and economic dataset (social network, mobile phone, satellites) that would allow estimation of the SSVI over finer geographical resolution to better represent the cyclone flooding vulnerability.

**7 Conclusion**

In this article, we defined vulnerability to cyclonic flooding in coastal Bangladesh based on the scientific literature publications (2007-2020). This entitles both to test scientifically the research procedure to assess the socio-spatial vulnerability and to state the coastal places in Bangladesh which are under studies or which are neglected , in order to steer forward researches where need is identified for coastal population. Therefore, the mouth of the Ganges appears to be a relatively little studied area, although it is very exposed to cyclonic flooding and vulnerable due to the density of the population (Chandpur and, to a lesser extent, Shariatpur).

Thus, we argue a better knowledge on the socio-spatial vulnerability can help sensitive classical cyclone flooding vulnerability analysis that tends to consider those groups or communities are equiprobably exposed to the hazard in a given spatiality. Consequently, we define a 'socio-spatial vulnerability index' (SSVI), as a function of both the probability of the cyclonic flood hazard and the sensitivity of delta inhabitants. Through this index, we quantify and map levels of this vulnerability. SSVI is also a new methodology to highlight socio-spatial inequalities between the coastal districts of Bangladesh by considering social, physical and infrastructural aspects of

vulnerability. The concept is thought in a systemic and interdisciplinary framework, considering these vulnerability aspects from the social and natural sciences and their interactions in-between. With this index and through its different components, we can better reveal why some groups or communities, at the districts level in the case of our study, are more vulnerable to cyclonic flooding, and could also be slower to recover than others. At final, we identify the most vulnerable places that are understudied, leading to consider differently the coastal area for the State, NGOs and OIs to prioritize their Preparedness and Recovery programs. Climate change could increase the intensity and frequency of tropical cyclones in the Bay of Bengal. It is therefore necessary accessing to appropriate cyclonic flood hazard from hydrological modelling efficient tools, in order to predict the future socio-spatial vulnerability patterns.

*Data availability.* https://zenodo.org/badge/DOI/10.5281/zenodo.5997080.svg

*Author contributions.* Conceptualization: A.B., M.B., N.L.; Methodology: A.B., M.B., N.L.; Software and formal analysis: A.B. N.L., M.B.; Writing: A.B., M.B., N.L., Review and editing: J.K., S.F.

*Competing interests.* The authors declare that they have not conflict of interest.

**Acknowledgments**

This work was financially supported by the French research agency (Agence Nationale de la Recherche; ANR) under the DELTA project (ANR-17-CE03-0001). We acknowledge financial support from CNES (through the TOSCA project BANDINO) and Embassy of France in Bangladesh. The CERG-C Program of the University of Geneva is thanked for providing the expertise and the framework to carry out this project.

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

| keywords in the title or abstract | Key words relating to flood disasters [hazard] | | Key words relating to social vulnerability [outcome] |
|---|---|---|---|
| **Words used for the research** | Coastal flood*/-ing | | (Social) *vulnerability* (index / assessment) |
| | Sea-level rise | **AND** | Human exposure |
| | Storm surge | | Coping* (capacities) |
| | Cyclonic / tropical *storm | | **Key words relating to the area of study** |
| | Disaster risk reduction | | |
| | | | Brahmaputra delta |
| | | | Bangladesh |
| **Search engines used** | Google Scholar, Istex, Scopus, Science Direct | | |
| **Limits** | [Date of Publication: 2007–2019; Abstract Available; English Language; Peer-reviewed articles] | | |

Alderman, K., Turner, L. R. and Tong, S.: Floods and human health: a systematic review, Environment 965 international, 47, 37–47, 2012.

**Appendix A2: Selected articles for the review**

| Search engines | Total selected articles | Significant articles | Non-significant or redundant |
|---|---|---|---|
| Google Scholar | 66 | 28 | 38 |
| Istex | 39 | 10 | 29 |
| Scopus | 24 | 6 | 18 |
| Science direct | 21 | 5 | 16 |

| Total | 149 | 49 | 101 |
|---|---|---|---|
| **Total selected for analysis** | | **49** | |

**Appendix A3: References of the articles used for the review analysis**

1. Abdullah, A. N. M., Zander, K. K., Myers, B., Stacey, N. and Garnett, S. T.: A short-term decrease in household income inequality in the Sundarbans, Bangladesh, following Cyclone Aila, Natural Hazards, 83(2), 1103–1123, 2016.

2. Ahamed, F., Hossain, M. Y., Fulanda, B., Ahmed, Z. F. and Ohtomi, J.: Indiscriminate exploitation of wild prawn postlarvae in the coastal region of Bangladesh: A threat to the fisheries resources, community livelihoods and biodiversity, Ocean & coastal management, 66, 56–62, 2012.

3. Ahmed, B., Kelman, I., Fehr, H. and Saha, M.: Community resilience to cyclone disasters in coastal Bangladesh, Sustainability, 8(8), 805, 2016.

4. Ahmed, N. and Diana, J. S.: Threatening "white gold": impacts of climate change on shrimp farming in coastal Bangladesh, Ocean & Coastal Management, 114, 42–52, 2015.

5. Ahmed, N., Allison, E. H. and Muir, J. F.: Rice fields to prawn farms: a blue revolution in southwest Bangladesh?, Aquaculture International, 18(4), 555–574, 2010.

6. Ahsan, M. N.: Can Strategies to Cope with Hazard Shocks be Explained by At-Risk Households' Socioeconomic Asset Profile? Evidence from Tropical Cyclone-Prone Coastal Bangladesh, International Journal of Disaster Risk Science, 8(1), 46–63, 2017.

7. Ahsan, M. N. and Warner, J.: The socioeconomic vulnerability index: A pragmatic approach for assessing climate change led risks–A case study in the south-western coastal Bangladesh, International Journal of Disaster Risk
Reduction, 8, 32–49, 2014.

8. Akter, S. and Mallick, B.: The poverty–vulnerability–resilience nexus: Evidence from Bangladesh, Ecological Economics, 96, 114–124, 2013.

9. Alam, A., Sammonds, P. and Ahmed, B.: Cyclone risk assessment of the Cox's Bazar district and Rohingya refugee camps in southeast Bangladesh, Science of The Total Environment, 704, 135360,
https://doi.org/10.1016/j.scitotenv.2019.135360, 2020.

10. Alam, E. and Collins, A. E.: Cyclone disaster vulnerability and response experiences in coastal Bangladesh, Disasters, 34(4), 931–954, 2010.

11. Alam, M. S., Sasaki, N. and Datta, A.: Waterlogging, crop damage and adaptation interventions in the coastal region of Bangladesh: A perception analysis of local people, Environmental Development, 23, 22–32, 2017.

12. Das, S., Hazra, S., Haque, A., Rahman, M., Nicholls, R. J., Ghosh, A., Ghosh, T., Salehin, M. and Safra de Campos, R.: Social vulnerability to environmental hazards in the Ganges-Brahmaputra-Meghna delta, India and Bangladesh, International Journal of Disaster Risk Reduction, 53, 101983, https://doi.org/10.1016/j.ijdrr.2020.101983, 2020.

13. Deb, A. K. and Haque, C. E.: 'Sufferings Start from the Mothers' Womb': Vulnerabilities and Livelihood War of the Small-Scale Fishers of Bangladesh, Sustainability, 3(12), 2500–2527, https://doi.org/10.3390/su3122500, 2011.

14. Garai, J.: Qualitative analysis of coping strategies of cyclone disaster in coastal area of Bangladesh, Natural Hazards, 85(1), 425–435, 2017.

15. Hossain, M. N.: Analysis of human vulnerability to cyclones and storm surges based on influencing physical and socioeconomic factors: evidences from coastal Bangladesh, International journal of disaster risk reduction, 13, 66–75, 2015.

16. Ishtiaque, A., Sangwan, N. and Yu, D. J.: Robust-yet-fragile nature of partly engineered social-ecological systems: a case study of coastal Bangladesh, Ecology and Society, 22(3), 2017.

17. Ishtiaque, A., Eakin, H., Chhetri, N., Myint, S. W., Dewan, A. and Kamruzzaman, M.: Examination of coastal vulnerability framings at multiple levels of governance using spatial MCDA approach, Ocean & Coastal Management, 171, 66–79, https://doi.org/10.1016/j.ocecoaman.2019.01.020, 2019.

18. Islam, M. A., Shitangsu, P. K. and Hassan, M. Z.: Agricultural vulnerability in Bangladesh to climate change induced sea level rise and options for adaptation: a study of a coastal Upazila, Journal of Agriculture and Environment for International Development (JAEID), 109(1), 19–39, https://doi.org/10.12895/jaeid.20151.218, 2015.

19. Islam, M. M., Sallu, S., Hubacek, K. and Paavola, J.: Limits and barriers to adaptation to climate variability and change in Bangladeshi coastal fishing communities, Marine Policy, 43, 208–216, 2014a.

20. Islam, M. M., Sallu, S., Hubacek, K. and Paavola, J.: Vulnerability of fishery-based livelihoods to the impacts of climate variability and change: insights from coastal Bangladesh, Regional Environmental Change, 14(1), 281–294, 2014b.

21. Islam, R. and Walkerden, G.: How bonding and bridging networks contribute to disaster resilience and recovery on the Bangladeshi coast, International Journal of Disaster Risk Reduction, 10, 281–291, 1020    https://doi.org/10.1016/j.ijdrr.2014.09.016, 2014.

22. Islam, R. and Walkerden, G.: How do links between households and NGOs promote disaster resilience and recovery?: A case study of linking social networks on the Bangladeshi coast, Natural Hazards, 78(3), 1707–1727, 2015.

23. Kulatunga, U., Wedawatta, G., Amaratunga, D. and Haigh, R.: Evaluation of vulnerability factors for cyclones: the case of Patuakhali, Bangladesh, International journal of disaster risk reduction, 9, 204–211, 2014.

24. Mallick, B. and Vogt, J.: Social supremacy and its role in local level disaster mitigation planning in Bangladesh, Disaster Prevention and Management: An International Journal, 20(5), 543–556, 2011.

25. Mallick, B. and Vogt, J.: Population displacement after cyclone and its consequences: Empirical evidence from coastal Bangladesh, Natural hazards, 73(2), 191–212, 2014.

26. Mallick, B., Rahaman, K. R. and Vogt, J.: Coastal livelihood and physical infrastructure in Bangladesh after cyclone 1030    Aila, Mitigation and Adaptation Strategies for Global Change, 16(6), 629–648, 2011.

27. Mallick, B., Ahmed, B. and Vogt, J.: Living with the risks of cyclone disasters in the south-western coastal region of Bangladesh, Environments, 4(1), 13, 2017.

28. Mallick, B. J., Witte, S. M., Sarkar, R., Mahboob, A. S. and Vogt, J.: Local adaptation strategies of a coastal community during cyclone Sidr and their vulnerability analysis for sustainable disaster mitigation planning in 1035    Bangladesh, Journal of Bangladesh Institute of Planners, 2, 158–168, 2009.

29. Miah, J., Hossain, K. T., Hossain, M. A. and Najia, S. I.: Assessing coastal vulnerability of Chittagong District, Bangladesh using geospatial techniques, J Coast Conserv, 24(6), 66, https://doi.org/10.1007/s11852-020-00784-2, 2020.

30. Mullick, Md. R. A., Tanim, A. H. and Islam, S. M. S.: Coastal vulnerability analysis of Bangladesh coast using fuzzy 1040    logic based geospatial techniques, Ocean & Coastal Management, 174, 154–169, https://doi.org/10.1016/j.ocecoaman.2019.03.010, 2019.

31. Parvin, G. A. and Shaw, R.: Microfinance institutions and a coastal community's disaster risk reduction, response, and recovery process: a case study of Hatiya, Bangladesh, Disasters, 37(1), 165–184, 2013.

32. Paul, B. G. and Vogl, C. R.: Impacts of shrimp farming in Bangladesh: challenges and alternatives, Ocean & Coastal 1045    Management, 54(3), 201–211, 2011.

33. Paul, B. K.: Why relatively fewer people died? The case of Bangladesh's Cyclone Sidr, Natural Hazards, 50(2), 289–304, 2009.

34. Paul, S. K. and Routray, J. K.: Household response to cyclone and induced surge in coastal Bangladesh: coping strategies and explanatory variables, Natural Hazards, 57(2), 477–499, 2011.

35. Pouliotte, J., Smit, B. and Westerhoff, L.: Adaptation and development: Livelihoods and climate change in Subarnabad, Bangladesh, Climate and development, 1(1), 31–46, 2009.

36. Quader, M., Khan, A. and Kervyn, M.: Assessing risks from cyclones for human lives and livelihoods in the coastal region of Bangladesh, International journal of environmental research and public health, 14(8), 831, 2017.

37. Rabby, Y. W., Hossain, M. B. and Hasan, M. U.: Social vulnerability in the coastal region of Bangladesh: An investigation of social vulnerability index and scalar change effects, International Journal of Disaster Risk Reduction, 41, 101329, https://doi.org/10.1016/j.ijdrr.2019.101329, 2019.

38. Rahman, M. K., Paul, B. K., Curtis, A. and Schmidlin, T. W.: Linking Coastal Disasters and Migration: A Case Study of Kutubdia Island, Bangladesh, The Professional Geographer, 67(2), 218–228, https://doi.org/10.1080/00330124.2014.922020, 2015.

39. Rakib, M. A., Sasaki, J., Pal, S., Newaz, Md. A., Bodrud-Doza, Md. and Bhuiyan, M. A. H.: An investigation of coastal vulnerability and internal consistency of local perceptions under climate change risk in the southwest part of Bangladesh, Journal of Environmental Management, 231, 419–428, https://doi.org/10.1016/j.jenvman.2018.10.054, 2019.

40. Roy, D. C. and Blaschke, T.: Spatial vulnerability assessment of floods in the coastal regions of Bangladesh, Geomatics, Natural Hazards and Risk, 6(1), 21–44, https://doi.org/10.1080/19475705.2013.816785, 2015.

41. Saha, C. K.: Dynamics of disaster-induced risk in southwestern coastal Bangladesh: an analysis on tropical Cyclone Aila 2009, Natural Hazards, 75(1), 727–754, 2015.

42. Sarkar, R. and Vogt, J.: Drinking water vulnerability in rural coastal areas of Bangladesh during and after natural extreme events, International Journal of Disaster Risk Reduction, 14, 411–423, 2015.

43. Saroar, M. and Routray, J. K.: In situ adaptation against sea level rise (SLR) in Bangladesh: does awareness matter?, International Journal of Climate Change Strategies and Management, 2(3), 321–345, 2010.

44. Saroar, M. M. and Routray, J. K.: Impacts of climatic disasters in coastal Bangladesh: why does private adaptive capacity differ?, Reg Environ Change, 12(1), 169–190, https://doi.org/10.1007/s10113-011-0247-4, 2012.

45. Shameem, M. I. M., Momtaz, S. and Rauscher, R.: Vulnerability of rural livelihoods to multiple stressors: A case study from the southwest coastal region of Bangladesh, Ocean & Coastal Management, 102, 79–87, 2014.

46. Sultana, F.: Living in hazardous waterscapes: Gendered vulnerabilities and experiences of floods and disasters, Environmental Hazards, 9(1), 43–53, 2010.

47. Swapan, M. S. H., Ashikuzzaman, M. and Iftekhar, M. S.: Dynamics of Urban Disaster Risk Paradigm: Looking Through the Perceived Lens of the Residents of Informal Settlements in Khulna City, Bangladesh, Environment and Urbanization ASIA, 11(1), 51–77, https://doi.org/10.1177/0975425320906269, 2020.

48. Uddin, M. N., Islam, A. S., Bala, S. K., Islam, G. T., Adhikary, S., Saha, D., Haque, S., Fahad, M. G. R. and Akter, R.: Mapping of climate vulnerability of the coastal region of Bangladesh using principal component analysis, Applied geography, 102, 47–57, 2019.

49. Younus, M. A. F. and Sharna, S. S.: Combination of community-based vulnerability and adaptation to storm surges in coastal regions of Bangladesh, Journal of Environmental Assessment Policy and Management, 16(04), 1450036, 2014.

**Appendix A4: Guiding question details**

1) **[WHO]**

Although inclusion criteria established the research only upon peer-reviewed literature, it is specified if they are academic or non-university researchers by mentioning their home institution. The disciplinary backgrounds give the academic outlook of the authors involved in the research, and is defined according to the author's affiliated working department or faculty.

2) **[WHERE]**

The spatial scales address the locations where the vulnerability assessments were made across Bangladesh. Firstly, they report if the assessment takes place in rural, urban or a mixed environment. Secondly, they specify the scales:

divisions, namely a region or, Zila which are districts; Upazila (thana), namely sub-districts, or unions, which are a gathering of two-three municipalities, or finally, if the analysis takes place at the household scale.

3) **[WHEN]**

The temporal scales of assessment refer to the hazard. The authors either consider a short- term period by focusing on a specific event like a cyclone, or a longer period, which could encompass a wider phenomenon, such as sea-level rise.

4) **[HOW]**

The theoretical approach used by the author should reveal the definition given to the vulnerability. For example, some authors refer to IPCC 2007's definitions or to 2014's definitions. Methods to assess vulnerability refers to the study design used by the authors. We distinguish quantitative (indicator-based, spatialized data, impact modelling), qualitative (interviews, participatory approaches, open surveys) and mixed methods (including both index and qualitative case studies).

5) **[WHAT]**

The scheme of the vulnerability categories encompassed a broad definition of vulnerability, including eventually a reference to what the authors could mean by social, human or people's vulnerability. We distinguish then between aspects and definitions of physical and systemic vulnerabilities.

**Appendix B: Citation frequency in the literature review of the selected variables (B.a) and coastal districts (B.b).**

| B.a | Total | % (n=49 studies) | B.b | Total | % (n=49 studies) |
|---|---|---|---|---|---|
| Education | 23 | 47% | Khulna | 18 | 37% |
| Shelters | 22 | 45% | Satkhira | 17 | 35% |
| Female | 20 | 41% | Barguna | 13 | 27% |
| Houses materials | 19 | 39% | Cox's Bazar | 12 | 24% |
| Dykes - embankments | 19 | 39% | Bagerhat | 11 | 22% |
| Poverty | 17 | 35% | Patuakhali | 11 | 22% |
| Drinking water | 16 | 33% | Pirojpur | 7 | 14% |
| Road | 14 | 29% | Chittagong | 7 | 14% |
| Children | 13 | 27% | Barisal | 7 | 14% |
| Population density | 12 | 24% | Noakhali | 6 | 12% |
| Sanitary access | 10 | 20% | Jhalokati | 5 | 10% |
| Disability | 9 | 18% | Bhola | 5 | 10% |
| Electricity access | 8 | 16% | Lakshmipur | 5 | 10% |
| Mobile access | 8 | 16% | Feni | 5 | 10% |
| | | | Chandpur | 4 | 8% |
| Elderly | 7 | 14% | Shariatpur | 2 | 4% |
| Hospital | 7 | 14% | | | |

| Minorities | 6 | 12% |
|---|---|---|

**Appendix C: Socio-spatial Vulnerability Index (SSVI) to cyclonic flooding methodology**

The Socio-spatial Vulnerability Index (SSVI) database comprises 17 variables, 4 themes and 16 districts. The calculation of SSVI is based on the methodology developed in Flanagan et al. (2011) and presented below:

Tier 1: For each of the 17 variables (*Poverty, Education (above 10 years of school), Age 14 or Younger, Female rate, Age 60 or Older, Disability, Religion, Houses made of organic materials, Unpaved roads, Access to electricity, Access to mobile, Access to sanitary, Unsafe drinking water, Hospitals, Dykes and embankments, Shelter capacity and Population density exposed to flooding*):

Rank the values from lowest to highest vulnerability degree

Calculate percentile rank (PR) for each district: Percentile Rank = (Rank-1)/(N-1), where N is the number of districts. The district with the highest vulnerability degree for a specific variable will have a PR score of 1, and the lowest vulnerability degree will have a PR score of 0.

Tier 2: For each of the 4 thematic domain *(Socioeconomic, Household Composition and Disability, Housing and Infrastructures and Cyclone Protections and Exposure)*:

Sum the PR of variables by domain

Rank the values from lowest to highest vulnerability degree

Calculate PR for each district

Tier 3: Overall SSVI

Sum the PR of the 4 thematic domains

Rank the values from lowest to highest vulnerability degree

Calculate PR for each district

Example of interpretation: If a district is in the 80th percentile (0.80 ranking) for the SVI, then 80% of the districts are either below or equal to that particular district regards to SSVI.

**Appendix D: Map layer' sources and description used to cartography**

| Map layer description | Details | Sources |
|---|---|---|
| Bay of Bengal map | Delimitation Bay of Bengal | Marine Gazetteer Placedetails. Indian Ocean/ IHO Sea Area. marineregions.org |
| Bangladesh map | Administrative units: borders (admin_0), divisions, districts, upazilas (admin_4) | DIVA GIS DATA |

| | | |
|---|---|---|
| Ganges-Brahmaputra-Meghna rivers | Major rivers in Bangladesh | Humdata, OCHA |
| Sundarbans map | Delimitation Sundarbans | ZHANG Haiying,NIU Zhenguo,LIU Chuang,SHI Ruixiang.Sundarbans Wetland in 2015 (Bangladesh) [DB/OL].Global Change Research Data Publishing & Repository, 2015. DOI:10.3974/geodb.2015.01.14.V1. / http://www.geodoi.ac.cn/WebEn/doi.aspx?Id=214 |
| Hospital location | Health sites in Bangladesh | Humdata, healthsites.io |
| Population density | Predicted number of people per square kilometer, represented by pixel. Estimated using the random forest (RF) model as described in Stevens, et al. (2015) | The most recent data were found on WorldPop (2015), aware that the data could differ from the reality, this source remains currently the most recent data available to map this density. |
| Coastal flooding map | For a 50 years-return period | Made by Jamal Khan, model (Khan et al. 2019) |
| Embankments location | Coastal embankments in Bangladesh | Bangladesh Water Development Board |
| Cyclone shelters location | Points of shelters in coastal Bangladesh | geodash.gov.bd |
| Level of poverty | Percentage of people living under the poverty line | Zila level povmap estimates, 2010 from The World Bank, World Food Programme and BBS |
| | Total of poor comprising both lower and upper the poverty line | |
| Paved and unpaved road | Paved or unpaved road | District wise Length of Road by Road Classification in 2018 under RHD (in KM), p.258 from Statistical Year Book Bangladesh 2018, 38TH EDITION |
| Cyclones shelters | shapefile inventories 3777 shelters to protect the coastal population from the cyclones and their capacity | available on geodash.gov |
| Religious minorities | Indicator aggregates religions such as Buddhist, Hindu, Christian and other. The official statistics inventory migration from urban areas to urban and rural areas on the one hand, from rural areas to rural and urban places on the other hand. | Aggregated from BBS Online Census 2014 |
| Unsafe water | The takes into account all spring of water that are not coming from the tap or from tube wells, they can come from simple wells, canal rivers and ponds. | Aggregated from BBS Online Census 2014 |
| Education Level | More than 10 years of school | BBS Online Census 2014 |
| Age years group | Addition of categories between ages 0-14 years | |
| | Addition of categories above ages 60 years | |
| Disable | Total disable people per district | |
| Sex | Gender (male or female) | |
| Type of structure (houses material) | House materials: JHUPRI, KACHA TINSHAD, SEMI-PUCCA, PUCCA, OTHERS_houses structure, Weak housing | |

| | | |
|---|---|---|
| Toilet facility | Access to toilet facilities | |
| Mobile | Access to mobile | |