# Peer review of "Bangladesh's vulnerability to cyclonic coastal flooding"

_Natural Hazards and Earth System Sciences, 2021_

## Author Comment (AC1)

**MS No.: nhess-2021-8**

**Response to review comments by Anonymous Referee #1 (R1)**

General comments:

R1.1 This review article aims to develop a metric to define coastal vulnerability induced by cyclonic hazards in Bangladesh. The vulnerability was defined in terms of physical infrastructure, social and cultural factors, and economic factors. The manuscript is well-written in general. However, the results obtained in this study are nothing special from the nature of coastal Bangladesh, mostly known from various studies. I have a few major observations which I would like the authors to address diligently before it can be considered for publication. Of particular concern is the motivation for carrying out such a study, as well as, its contribution to the existing scientific literature.

> Authors' response: Literature on cyclonic flooding vulnerability along the coast of Bangladesh typically examines either physical or social characteristics of the vulnerability to this hazard. This paper takes slightly different approach: it argues the need to consider the socio-spatial specific context which transcends the classical social and economic responses to integrate physical and infrastructural conditions as a basis of understanding and addressing cyclonic flooding vulnerabilities.

> First, it sets out the framework in which cyclonic flooding vulnerability should be placed and shows the issue tied to its indicators. Indeed, majority of authors does not explain really the conceptual framework underlying their vulnerability assessment, making more difficult to interpret and re-use their results for scientists and stakeholders.

> Moreover, lots of studies used a large set of indicators (i.e., Quader et al. 2017), which are reduced to smaller uncorrelated factors set using statistical methods, such as the principal components analysis, it raises as a consequence complex issues of number of principal components to retain, their significance, and the normalization choice, among other. Contrariwise, we analyzed the relevant indicators related to the cyclonic flooding vulnerability based on expert knowledge from a strict literature review. The literature review is a rich source to understand the main causes, translated as indicators, of social-spatial vulnerability to cyclonic flooding, their relative importance and interactions. Improved incorporation of the regional context helps to select significant indicators, at total 17 in our case, that not only reflect vulnerability as a state (e.g. poverty, age), but also as a situation (e.g. house type, shelter capacity).

> Finally, an overview of the conceptualization of socio-spatial cyclonic flooding vulnerability is presented thought the SSVI. It is a simple computation of index which has a more understandable construction and appears easier to replicate for decision makers. Furthermore, unlike other studies which have developed physical or social cyclonic flooding vulnerability assessment, in our study, cyclone hazard flooding has been considered and incorporated in the social vulnerability model (i.e. SSVI).

> This paper contributes to bring a paradigm shift, considering the integrated socio-spatial vulnerability, and new perspective to assess the vulnerability to cyclonic coastal flooding. It provides a framework for future research on the topic of integrated socio-spatial vulnerability assessments to cyclonic floods, as a basis for adaptation strategies.

Specific comments:

R1.2 I am surprised to see that several important recent studies were not included in this study, given a systematic review was done.

> Authors' response: To be precise, we did not conduct a 'Systematic Review' process. This term was wrongly used in the titles of the Appendix A2 and A3, this has been corrected now. This work goes further than a 'Systematic Review', because in addition to assess the vulnerability indicators to cyclonic

flooding drawing on literature, we define a new metric, called 'socio-spatial vulnerability index' (SSVI), as function of both the probability of the cyclone flood hazard and the sensitivity of inhabitants.

Considering, the review process used to collect articles, as mentioned section 4.1, only the articles responding to the following search sequence, in the electronic databases, were considered: ("coastal flood*" OR "sea-level rise" OR "Storm surge" OR "Cyclonic storm" OR "Disaster risk reduction") AND ("Bangladesh" OR "Brahmaputra delta") AND ("*vulnerability" OR "human exposure" OR "coping*").

R1.3 A few examples are given below. I feel the incorporation of these studies will strengthen the motivation and validity of this article:

https://doi.org/10.5194/nhess-19-353-2019 (Islam, Md. F., Bhattacharya, B., Popescu, I. (2019). Flood risk assessment due to cyclone-induced dike breaching in coastal areas of Bangladesh. Natural Hazards and Earth System Sciences, 19, 353-368)

Authors' response: The main objective of Islam et al. (2019) is to estimate the exposed areas to storm surge flood of the polder 48 (Patuakhali district), using a hydrodynamic model under different scenarios in a context of climate change (tide, sea level, dike breach and cyclone landfall angle). This article focuses on flooding, i.e. on the hazard, and not on the population/territory vulnerability to cyclonic flooding. This article was not therefore considered by our literature review process.

https://doi.org/10.1016/j.scitotenv.2019.05.048 (Adnan, M. S. G., Haque, A., Hall, J.W. (2019). Have coastal embankments reduced flooding in Bangladesh?. Science of the Total Environment, 682, 405-416.)

Authors' response: Adnan et al. (2019) analyzed the beneficial and harmful impacts of embankment construction on the territories. They focused on pluvial, fluvial-tidal and cyclonic floods by comparing observed and modelled floods, with and without polders. Islam et al. (2019) article focuses on flooding, i.e. on the hazard, and not on the population/territory vulnerability to cyclonic flooding. This article was not therefore considered by our literature review process.

https://doi.org/10.1016/j.landusepol.2020.104868 (Adnan, M. S. G., Abdullah, A. Y. M., Dewan, A., & Hall, J. W. (2020). The effects of changing land use and flood hazard on poverty in coastal Bangladesh. Land Use Policy, 99, 104868.)

Authors' response: Adnan et al. (2020) were interested in pluvial flood risk, as a function of different land use/land cover scenarios, in five coastal districts: Bagerhat, Jessore, Khulna, Pirojpur, and Satkhira. Although this article answered to the keywords search query, it was discarded because it did not focus on the main issue of our study, i.e. the socio-spatial vulnerability to the cyclonic flood. An interesting point is raised by this work about the influence of the land use/land cover scenarios on the flood dynamics. We note this information is already considered as an input parameter, named soil roughness, in the hydrodynamical modelling schema (Khan et al. 2019) used in our study (section 3).

https://doi.org/10.1007/978-3-319-71093-8_16 (Haque A., Kay S., Nicholls R.J. (2018) Present and Future Fluvial, Tidal and Storm Surge Flooding in Coastal Bangladesh. In: Nicholls R., Hutton C., Adger W., Hanson S., Rahman M., Salehin M. (eds) Ecosystem Services for Well-Being in Deltas. Palgrave Macmillan, Cham.)

Authors' response: Although the book chapter of Haque et al. (2018) might be relevant to the physical characterization of the flood type (fluvial, tidal, fluvio-tidal, storm surge floods) in the GBM delta, it does not deal with the population/territory vulnerability to cyclonic flooding, that is the main issue of our study. Therefore, this paper which fails to meet our search requirements, was not considered in our literature

review process. But, following your recommendation, we propose to add this reference in the new paragraph in the introduction about the existing flood types in the GBM delta

https://doi.org/10.1007/s11069-017-3027-8 (Younus Md. A. F. (2017). An assessment of vulnerability and adaptation to cyclones through impact assessment guidelines: a bottom-up case study from Bangladesh coast, 89, 1437-1459.)

Authors' response: Younus et al. (2017) propose an assessment of vulnerability based on a qualitative approach (by participatory rapid appraisal with semi-structured questions and groups of discussions) with the population of the Bawalkor village (Barguna district). Several high vulnerability issues are identified like seed-bed damage, primary occupation loss or culture fishpond loss. A second part of this study concerns adaptation issues (need to reconstruct the shelter, need of relief from the interest of loan for 1 year after the disaster, need to reconstruct roads, and so on). While interesting in its approach, this article does not identify key variables defining socio-spatial vulnerability to cyclonic flooding at the district level based on variables available in public databases. It was therefore not retained through our literature review process.

https://doi.org/10.1007/s41748-018-0034-1 (Hossain et Paul (2018). Vulnerability Factors and Effectiveness of Disaster Mitigation Measures in the Bangladesh Coast. Earth Systems and Environment, 2: 55–65.)

Authors' response: Hossain et Paul (2018) propose a review about the effectiveness of current disaster mitigation measures undertaken by individuals, GOs and NGOs to minimize the cyclone impacts in coastal areas (i.e. Gabura in Satkhira district and Golkhali in Khulna district). Although this article brings a very interesting contribution into the topic, it does not provide more information on population/territory vulnerability to cyclonic flooding than those previously published by the same authors, i.e. Hossain, (2015) and Paul and Routray (2010), and already considered in our literature review process. This article was not therefore considered by our literature review process.

R1.4 The coastal region of Bangladesh is subject to multiple types of flooding such as pluvial, fluvio-tidal, and cyclone-induced storm surge flooding. Cyclonic flood events in this region are usually accompanied by pluvial flooding. Flood vulnerability tends to change with an increase in the complexities of flood events. In this manuscript, the authors have not discussed the complex coastal flooding processes and associated vulnerability in Bangladesh. They could consult some of the articles suggested above.

Authors' response: We agree with your comment, insofar as in the coastal region of Bangladesh, multiple physical processes are responsible for flooding at various spatial and temporal scales - including pluvial, fluvio-tidal, and cyclone-induced storm surge flooding (e.g., Haque et al. 2018). Fortunately, the cyclone-induced storm surges are predominantly during pre- and post-monsoon seasons, when the flow at Ganges-Brahmaputra river system is relatively low.

Fluvial floods, on the other hand, are concentrated during summer monsoon spells, predominantly affecting the floodplains of Brahmaputra. Possible aggravation of the fluvial flood with its coincidence with spring tide has been acknowledged (e.g., the 1998 fluvial-tidal flood case was examined in Adnan et al. 2019). At the same time, co-incidence of Ganges and Brahmaputra flood peak is also noted as an underlying reason for the flood in 1998, as well as the 1988 flood (e.g. Mirza 2002). Evidently, there is still a research gap that needs to be addressed on this complex coastal setting regarding the fluvio-tidal floods. We have a similar remark for pluvial flooding, that accompanies the cyclonic events (Uddin et al. 2019). The flooding that accompanies this cyclonic/post-cyclonic precipitation is challenging. First, due to uncertainty in the rainfall estimate from model as well as satellite over this region (e.g., Rahman

et al. 2011), and second, due to uncertainty in the inland topography. The same topography issue persists for modelling inland inundation from storm surge.

In order to better appreciate this flood complexity and diversity in the coastal region of Bangladesh, we propose to add these sentences in the introduction section:

"These cyclone-induced storm surges are more frequent during pre-monsoon (May) and post-monsoon (October–November) season, when the flow at Ganges-Brahmaputra river system is relatively low and the rainfall moderate (Uddin et al. 2019), thereby reducing the likelihood of experiencing compound events, i.e. combined with other flood types as pluvial and fluvial (Haque et al. 2018). Concerning, the contribution of the heavy rainfall from landfalling cyclones to the floods, in addition to the storm surge, it still remains unexplored for this region. Therefore, hereafter, the study focused only on the cyclone-induced storm surge flood hazard".

*Haque, A., Kay, S. and Nicholls, R.J., (2018). Present and future fluvial, tidal and storm surge flooding in coastal Bangladesh. In Ecosystem Services for Well-Being in Deltas (pp. 293-314). Palgrave Macmillan, Cham.*

*Uddin, M., Li, Y., Cheung, K. K., Nasrin, Z. M., Wang, H., Wang, L., & Gao, Z. (2019). Rainfall contribution of Tropical Cyclones in the Bay of Bengal between 1998 and 2016 using TRMM satellite data. Atmosphere, 10(11), 699.*

*Adnan, M. S. G., Haque, A., & Hall, J. W. (2019). Have coastal embankments reduced flooding in Bangladesh?. Science of the total environment, 682, 405-416.*

*Mirza, M. M. Q. (2002). Global warming and changes in the probability of occurrence of floods in Bangladesh and implications. Global environmental change, 12(2), 127-138.*

*Rahman, M. M., Singh Arya, D., Goel, N. K., & Mitra, A. K. (2012). Rainfall statistics evaluation of ECMWF model and TRMM data over Bangladesh for flood related studies. Meteorological Applications, 19(4), 501-512.*

R1.5 Section 4.1: The methodology for selecting articles is not clear. First, I am not convinced with the inclusion of Google Scholar as one of the search engines as it tends to include articles that are not scientifically valid. How did the authors ensure that whether articles obtained from Google scholar are peer-reviewed or not?

Authors' response: First of all, we should call to mind that we used four traditional academic citation databases: ISTEX, Science Direct, Scopus and Web of Science. In addition, we chose to complete our analysis with the literature extracted from Google Scholar (GS) database. There are several reasons for this choice: GS is free of use (which is not the case of Scopus and Web of Science); and although it was shown that the majority of literature identified using Web of Science was also found using GS (Haddaway et al. 2015), it was also demonstrated that a large fraction (9–30%) of highly-cited documents in the Social Sciences and Humanities could be invisible to Web of Science and Scopus (Martin-Martin et al. 2018). Here, our primary area of concern is the vulnerability of the population to cyclonic flooding, therefore we thought it was important to complete this classical literature analysis with also the results from GS database.

We propose to modify and add this sentence in the main text to justify the use of GS database search engine (section 4.1) :

"Four traditional academic citation databases: ISTEX, Science Direct, Scopus and Web of Science were used for the literature review process. In addition, we completed the analysis with the literature extracted from Google Scholar database. Despite it was shown that the majority of literature identified using Web of Science was also found using Google Scholar (Haddaway et al. 2015), it was also demonstrated that a large fraction (9–30%) of highly-cited documents in the Social Sciences and Humanities could be invisible to Web of Science and Scopus (Martin-Martin et al. 2018)."

Moreover, the table below presents each peer-review journal considered in our literature review process (title, ISSN, CiteScore, highest percentile, publisher). This information comes from the Scopus database).

Table 1

| Nb | Title | Type | Peer-Review | ISSN | CiteScore | % | Highest percentile (from Scopus) | | Country | Publisher |
|---|---|---|---|---|---|---|---|---|---|---|
| | | | | | | | Rank | Subject Area | | |
| 1 | Applied Geography | journal | yes | 1436228 | 6,4 | 95 | 29/679 | Geography, Planning and Development | Netherlands | Elsevier BV |
| 1 | Aquaculture International | journal | yes | 1573143X, 09676120 | 2,9 | 73 | 89/334 | Agronomy and Crop Science | Netherlands | Springer Netherlands |
| 1 | Climate and Development | journal | yes | 17565529, 17565537 | 4,1 | 88 | 27/239 | Development | United Kingdom | Taylor and Francis Ltd. |
| 1 | Disaster Prevention & Management | journal | yes | 9653562 | 2,1 | 62 | 105/275 | Health (social science) | United Kingdom | Emerald Group Publishing Ltd. |
| 2 | Disasters | journal | yes | 03613666, 14677717 | 3,2 | 88 | 30/249 | General Social Sciences | United Kingdom | Wiley-Blackwell Publishing Ltd |
| 1 | Ecological Economics | journal | yes | 9218009 | 6,9 | 93 | 44/637 | Economics and Econometrics | Netherlands | Elsevier |
| 1 | Ecology and Society | journal | yes | 17083087 | 7,5 | 93 | 25/370 | Ecology | Canada | The Resilience Alliance |
| 1 | Environment and Urbanization ASIA | journal | yes | 09763546, 09754253 | 1,1 | 56 | 88/200 | Urban Studies | United States | SAGE Publications Inc. |
| 1 | Environmental Development | journal | yes | 22114645 | 4,8 | 91 | 57/679 | Geography, Planning and Development | Netherlands | Elsevier BV |
| 1 | Environmental Hazards | journal | yes | 14642867, 17477891, 18780059 | 2,2 | 76 | 293/124 | Sociology and Political Science | United Kingdom | Taylor and Francis Ltd. |
| 1 | Environments | journal | yes | 20763298 | 0,7 | 27 | 152/210 | General Environmental Science | Switzerland | MDPI AG |
| 1 | Geomatics, Natural Hazards and Risk | journal | yes | 19475713, 19475705 | 5,2 | 89 | 20/187 | General Earth and Planetary Sciences | United Kingdom | Taylor and Francis Ltd. |
| 1 | International Journal of Climate Change Strategies and Management | journal | yes | 17568692 | 2,4 | 72 | 67/239 | Development | United Kingdom | Emerald Group Publishing Ltd. |
| 7 | International Journal of Disaster Risk Reduction | journal | yes | 22124209 | 4,4 | 89 | 9/795a | fety Research | United Kingdom | Elsevier Ltd. |
| 1 | International Journal of Disaster Risk Science | journal | yes | 21926395, 20950055 | 3,6 | 84 | 108/679 | Geography, Planning and Development | United States | Springer Science + Business Media |
| 1 | International Journal of Environmental Research and Public Health | journal | yes | 16617827, 16604601 | 3,0 | 66 | 174/516 | Public Health, Environmental and Occupational Health | Switzerland | MDPI Digital Publishing Institute |
| 1 | Journal of Agriculture and Environment for International Development | journal | yes | 22402802 | 0,8 | 43 | 114/203 | General Agricultural and Biological Sciences | Italy | Italian Agency for Development Co. |
| 1 | *Journal of Bangladesh Institute of Planners\*\*\** | *journal* | *yes* | *20759363, 24088587* | *---* | *---* | *---* | *---* | *Bangladesh* | *Bangladesh Academy of Sciences* |
| 1 | Journal of Coastal Conservation | journal | yes | 18747841, 14000350 | 2,2 | 54 | 73/160 | Nature and Landscape Conservation | Netherlands | Springer Netherlands |
| 1 | Journal of Environmental Assessment Policy and Management | journal | yes | 14643332 | 1,6 | 56 | 293/679 | Geography, Planning and Development | Singapore | World Scientific Publishing Co. Pte Ltd |
| 1 | Journal of Environmental Management | journal | yes | 10958630, 03014797 | 7,6 | 95 | 15/333 | Management, Monitoring, Policy and Law | United States | Academic Press Inc. |
| 1 | Marine Policy | journal | yes | 0308597X | 5,3 | 97 | 19/685 | Law | United Kingdom | Elsevier Ltd. |
| 1 | Mitigation and Adaptation Strategies for Global Change | journal | yes | 15731596, 13812386 | 5,7 | 87 | 48/370 | Ecology | Netherlands | Springer Netherlands |
| 7 | Natural Hazards | journal | yes | 15730840, 0921030X | 5 | 87 | 28/217 | Water Science and Technology | Netherlands | Springer Netherlands |
| 6 | Ocean and Coastal Management | journal | yes | 9645691 | 4,3 | 79 | 46/219 | Aquatic Science | United Kingdom | Elsevier BV |
| 1 | Professional Geographer | journal | yes | 14679272, 00330124 | 3,1 | 79 | 142/679 | Geography, Planning and Development | United Kingdom | Taylor and Francis Ltd. |
| 2 | Regional Environmental Change | journal | yes | 14363798 | 6,5 | 76 | 20/83G | lobal and Planetary Change | Germany | Springer Verlag |
| 1 | Science of the Total Environment | journal | yes | 00489697, 18791026 | 8,6 | 92 | 10/132 | Environmental Engineering | Netherlands | Elsevier |
| 2 | Sustainability | journal | yes | 20711050 | 3,2 | 80 | 132/679 | Geography, Planning and Development | Switzerland | MDPI AG |

Nb: Number of items considered in our litterature review

ISSN: International Standard Serial Number

CiteScore (from Scopus): Measuring the average citations for a journal. This is calculated by the number of citations within the past three years divided by the number of all items published in the same years. The higher the CiteScore, the more valuable the journal is deemed to be.

\*\*\*: Journal not referenced in Scopus database

*Martín-Martín, A., Orduna-Malea, E. & Delgado López-Cózar, E. Coverage of highly-cited documents in Google Scholar, Web of Science, and Scopus: a multidisciplinary comparison. Scientometrics 116, 2175–2188 (2018). https://doi.org/10.1007/s11192-018-2820-9*

*Haddaway NR, Collins AM, Coughlin D, Kirk S (2015) The Role of Google Scholar in Evidence Reviews and Its Applicability to Grey Literature Searching. PLoS ONE 10(9): e0138237. https://doi.org/10.1371/journal.pone.0138237*

R1.6 The authors said that "This review excludes non-peer-reviewed articles, in the aim to obtain a state of the art's review of the current knowledge extracted from scientific literature only, to make sure of high scientific quality standards in the selected articles". How did they define "high scientific quality standards"? This comment does not apply to articles obtained from other search engines.

Authors' response: Only peer-reviewed articles (see Table 1) were considered as being of high scientific quality standards because this literature goes through a reviewing process, where studies are verified by the editor and other scientists, the reviewers. In those articles, a scientific methodology is employed to test empirically research hypotheses and to be sure that data gathering, treatment and results analysis are logical, verifiable and reproducible. Moreover, peer-reviewed articles were used in the aim to ease their access for everybody through search engines, excluding internal or not-online reports from private and non-governmental authors, that cannot be found again by other scientists.

We propose to add this sentence in the main text to explain what we mean by the "high scientific quality standards" (section 4.1):

"In those articles, a scientific methodology is employed to test empirically research hypotheses and to be sure that data gathering, treatment and results analysis are logical, verifiable and reproducible."

R1.7 I believe the authors found many articles from the initial search. How the 49 articles were selected? What were the exclusion criteria? The authors need to clarify in their manuscript. Appendix A1 only included the inclusion criteria.

Authors' response: We propose to add this paragraph in the main text to clarify our review process (section 4.1):

"The initial search used inclusion criteria to be certain the first range of articles selected focused on studies about vulnerability to cyclonic floods in Bangladesh. A second selection of the first range of articles screened documents one by one to check if they explored vulnerability to cyclones and if they specifically did focus on areas placed in coastal Bangladesh, and not vulnerability to river flood, for instance."

R1.8 Bangladesh has a long history of implementing various flood adaptation and prevention measures against various coastal flooding. Coastal embankments are the well-known flood protection measures that have been adopted in Bangladesh. Existing studies have quantified the effectiveness of such measures against complex coastal flooding including cyclonic flood events. This article significantly simplifies the actual coastal flood processes. The authors need to discuss how the adoption of various flood interventions alters flood vulnerability in the coastal region.

Authors' response: Please see our reply to point R1.4 concerning the flood complexity in coastal zone. The objective of our study is not to test the effectiveness of cyclone flooding defense measures but to propose a SSVI, based on a picture of the current situation into every district, in terms of dykes and embankments presence, shelter capacity and land cover, through the hydrodynamical modelling. Therefore, a discussion on the adoption of interventions is out of the scope of our study.

However, we propose to add this sentence and reference in the section 3:

"Conventional methods for reducing the effect of cyclonic flooding in this region are: embankments, polderization, coastal afforestation and shelter construction (Rahman et al. 2015)."

*Rahman, M. A., & Rahman, S. (2015). Natural and traditional defense mechanisms to reduce climate risks in coastal zones of Bangladesh. Weather and Climate Extremes, 7, 84-95.*

R1.9 Line 454-455: According to this study, the length of dykes and embankments is negatively correlated with cyclone vulnerability. But the following study provided various evidence indicating that such embankments were less effective against historical cyclonic flood events. Moreover, these measures promoted associated pluvial flooding. The authors need to justify the selection of various factors used in developing the vulnerability metric. https://doi.org/10.1016/j.scitotenv.2019.05.048

Authors' response: We remind here that the justification of each factors, used in the SSVI index computation, is drawn from our literature review process. Therefore, based on our literature review process, we hypothesized that the presence of dikes and embankments reduces the exposure of populations and territories located behind them. The variable dikes and embankments is cited in 39% of selected articles (i.e. 19 among 49 articles considered, see Appendix B) to define the vulnerability to cyclonic flooding. However, we agree with the Referee#1 about a need for discussion in the text on the possible negative effects that could be also caused by dikes and embankments. Therefore, we suggest to add this paragraph in the Discussion section to discuss this point:

"In recent studies, dike breaching is found to be the reason behind flooding (e.g., Hossain 2015, Adnan et al. 2019, Khan et al. under review). For example, Hossain (2015) mentioned that dikes and embankments may be in very poor condition and may not perform its protective function. Similar results are suggested by Khan et al. (under review) from analysis of the recent cyclone Amphan that made landfall in May 2020. Additionally, based on the results presented in Adnan et al. 2019, one might argue that the dikes were not adequate to protect against a certain event (either because of its design, or by gradual degradation of the dike). As noted in this article, for Sidr, if the dike has not had breached, the inundation would have been 18% of the coastal area, whereas the dike-breach increased the number to 35%. Unfortunately, presently, dike condition information is not available in public and accessible databases, and according to our understanding, not monitored either. The assumption in our

hydrodynamical modelling is that the dikes are in well serviced condition, e.g., dike breaching (which is a different geotechnical process, and far from the scope of this paper) was not modelled. Therefore, we did not include it in our methodology. However, there is no doubt that this issue about the balance between the negative and the positive contribution of dikes and embankments will have to be taken into account in future vulnerability to cyclonic flooding studies in this region."

*Hossain, M. N.: Analysis of human vulnerability to cyclones and storm surges based on influencing physical and socioeconomic factors: evidences from coastal Bangladesh, International journal of disaster risk reduction, 13, 66–75, 2015.*

*Adnan, M. S. G., Haque, A., & Hall, J. W. (2019). Have coastal embankments reduced flooding in Bangladesh?. Science of the total environment, 682, 405-416*

*Khan, M. J. U., Durand, F., Bertin, X., Testut, L., Krien, Y., Islam, A. K. M. S., Pezerat, M., and Hossain, S.: Towards an efficient storm surge and inundation forecasting system over the Bengal delta: Chasing the super-cyclone Amphan, Nat. Hazards Earth Syst. Sci. Discuss. [preprint], https://doi.org/10.5194/nhess-2020-340, in review, 2020.*

R1.10 Similarly, the capacity of shelters was perceived to be negatively associated with cyclone vulnerability. However, results from various studies indicated that various social factors are associated with the use of shelters during a cyclone event. During several events, people were reluctant to use cyclone shelters Please see the following study: https://doi.org/10.1111/disa.12062 (Mallick, B. (2014). Cyclone shelters and their locational suitability: An empirical analysis from coastal Bangladesh. *Disasters, 38*(3), 654-671.)

Such a scenario creates uncertainty in the obtained results. The authors should include a discussion on uncertainties related to the factors considered for the vulnerability metric.

Authors' response: We bring out here that the justification of each factors, used in the SSVI index computation, is drawn on our literature review process. Therefore, based on our literature review process, we hypothesized that the shelter capacity reduces the exposure of populations within a one-kilometer radius of it. The variable shelter capacity is cited in 45% of selected articles (i.e. 22 among 49 articles considered, see Appendix B) to define the vulnerability to cyclonic flooding.

However, we agree with you, as mentioned in the text lines 278-280, some studies pointed it out: "Proximity, poor hygienic condition (Kulatunga et al., 2014), lack in communication means and delivery of sanitation and drinking water (Saha, 2015) are some examples for additional reluctant aspects for many of the families."

A new sub-section "6.2 Representativeness and quality of the data" will be added to the Discussion in order to mention the limits of our study due mainly to the representativeness of the data.

"6.2 Representativeness and quality of the data

Some of the variables used to calculate the SSVI have some underlying assumptions that might differ from reality and may give a false representation of the situation on the field. For example, we assumed that dikes and embankments protect people and agricultural production, but everything depends on their condition, maintenance and breach presence (Younus and Sharna, 2014; Mullick et al., 2019; Hossain, 2015). As mentioned by these authors, dikes and embankments may be in very poor condition, but this information is not available in public and accessible databases, and not monitored to our knowledge. Therefore, we assume they play their full protective role through their physical presence. In addition, after a cyclone, populations can and do settle on dikes, sometimes the only free public spaces left to the people, and are therefore very exposed (Alam and Collins, 2010).

The shelter capacity is another example of variable that does not appear always representative of the situation on the field. It can be assumed that the presence of cyclone shelters reduces people's exposure and therefore their vulnerability. However, the presence of cyclone shelters in close proximity to homes, within a radius of 1 to 1.5km, does not mean that they are useful and used. Like mentioned by Mallick et al. 2017, shelters are not optimally placed on territories in to order be easily accessible and accommodate as many people as possible, but are rather situated near the supreme classes dwellings. These buildings are not always maintained and do not meet the requirements of the local society: men and women are mixed, there are no women-only sanitary facilities, and people may feel in insecurity (Kulatunga et al., 2014; Saha, 2015). Accessibility to cyclone shelters is also conditioned by the state of the roads. Therefore, we chose to represent the road condition by the variable "paved or unpaved road". However, we can consider this information is valid for estimating vulnerability only before the passage of a cyclone (in the prior-disaster phase) because then the roads can be very damaged as mentioned by Saroar and Routray (2010) during the rainy season."

The lines 505-511 will be shifted in the sub-section "6.3 Limits of the research"

R1.11 Section 6.1: This section lacks in critical discussion of the obtained results. The authors need to discuss how their findings are similar or different from the results of various existing studies.

Authors' response: The definition of vulnerability varies greatly from one study to another, making a comparison difficult between several studies. However, the one made by Quader et al. (2017) seems the most relevant to us in order to perform this exercise.

Vulnerability to cyclone risk has been extensively studied in Bangladesh. However, our study differs from the others on several points:
• its entry by the territory seems more useful to us in order to help decision makers in their decision making and their strategic orientations in risk management,
• a global and deductive approach was given to the definition of vulnerability (and not only for each of its dimensions),
• a robust probabilistic cyclonic flood hazard map was made, based on a dataset of 3600 statistically and physically consistent synthetic cyclone events.
• and finally, the definition of vulnerability alone was elaborated, without the adaptive capacities, but including exposure.

We suggest to complete the discussion section with these new paragraphs:

[revised manuscript text omitted]

R1.12 Finally, the motivation for carrying out such a study needs to be very clear. The authors mainly argued that "Thus, more than developing a new vulnerability concept, the novelty of our study lies in the methodological adaptation of existing approaches, like social vulnerability index (Flanagan et al., 2011), to the specific cyclone flooding context." I feel this is not enough. The authors should clearly identify the gaps in the existing literature and explain how this study addresses those gaps.

Authors' response: Please see our previous replies to points R1.1 and R1.11 concerning the novelty of the study.

---

## Author Comment (AC2)

**Response to review comments by Anonymous Referee #2 (R2)**

General comments:

R2.1    This article develops the socio-spatial vulnerability of 16 coastal districts of Bangladesh based on the vulnerability indicators at district levels. The contribution of this analysis is not new, as the authors have already shown in their literature review that a growing body of literature describes how vulnerable the coastal communities in Bangladesh are. In this paper, the specification that I see is the combination of people and place vulnerability to compare the districts, which are mostly known from various studies.

> Authors' response: Literature on cyclonic flooding vulnerability along the coast of Bangladesh typically examines either physical or social characteristics of the vulnerability to this hazard. This paper takes slightly different approach: it argues the need to consider the socio-spatial specific context which transcends the classical social and economic responses to integrate physical and infrastructural conditions as a basis of understanding and addressing cyclonic flooding vulnerabilities.

> First, it sets out the framework in which cyclonic flooding vulnerability should be placed and shows the issue tied to its indicators. Indeed, majority of authors does not explain really the conceptual framework underlying their vulnerability assessment, making more difficult to interpret and re-use their results for scientists and stakeholders.

> Moreover, lots of studies used a large set of indicators (Quader et al. 2017), which are reduced to smaller uncorrelated factors set using statistical methods, such as the principal components analysis, it raises as a consequence complex issues of number of principal components to retain, their significance, and the normalization choice, among other. Contrariwise, we analyzed the relevant indicators related to the cyclonic flooding vulnerability based on expert knowledge from a strict literature review. The literature review is a rich source to understand the main causes, translated as indicators, of social-spatial vulnerability to cyclonic flooding, their relative importance and interactions. Improved incorporation of the regional context helps to select significant indicators, at total 17 in our case, that not only reflect vulnerability as a state (e.g. poverty, age), but also as a situation (e.g. house type, shelter capacity).

> Finally, an overview of the conceptualization of socio-spatial cyclonic flooding vulnerability is presented thought the SSVI. It is a simple computation of index which has a more understandable construction and appears easier to replicate for decision makers. Furthermore, unlike other studies which have developed physical or social cyclonic flooding vulnerability assessment, in our study, cyclone hazard flooding has been considered and incorporated in the social vulnerability model (i.e. SSVI).

> This paper contributes to bring a paradigm shift, considering the integrated socio-spatial vulnerability, and new perspective to assess the vulnerability to cyclonic coastal flooding. It provides a framework for future research on the topic of integrated socio-spatial vulnerability assessments to cyclonic floods, as a basis for adaptation strategies.

Specific comments:

R2.2    I am a bit concerned about the literature search portal, like google is not such confident sources for academic publication searches. Then it is needed to describe how the final list of the review prepared, based on what criteria the article was selected for review, i.e. exclusion criteria must be written in the methodology section.

> Authors' response: We should remind here first of all that we used four traditional academic citation databases: ISTEX, Science Direct, Scopus and Web of Science. In addition, we chose to complete our analysis with the literature extracted from Google Scholar (GS) database. There are several reasons for

this choice: GS is free to use (which is not the case of Scopus and of Web of Science); and although it has been shown that the majority of literature identified using Web of Science was also found using GS (Haddaway et al. 2015), it has also demonstrated that a large fraction (9–30%) of highly-cited documents in the Social Sciences and Humanities could be invisible to Web of Science and Scopus (Martin-Martin et al. 2018). Here, our primary area of concern is the vulnerability of the population to cyclonic flooding, therefore we felt it was important to complete this classical literature analysis with also the results from GS database. (see also answer to comment R1.5 and R1.6).

We propose to modify and add this sentence in the main text to justify the use of GS database search engine (section 4.1 Methodology):

"Four traditional academic citation databases: ISTEX, Science Direct, Scopus and Web of Science were used for the literature review process. In addition, we completed the analysis with the literature extracted from Google Scholar database. Despite it has been shown that the majority of literature identified using Web of Science was also found using Google Scholar (Haddaway et al. 2015), it has also demonstrated that a large fraction (9–30%) of highly-cited documents in the Social Sciences and Humanities could be invisible to Web of Science and Scopus (Martin-Martin et al. 2018)."

*Martín-Martín, A., Orduna-Malea, E. & Delgado López-Cózar, E. Coverage of highly-cited documents in Google Scholar, Web of Science, and Scopus: a multidisciplinary comparison. Scientometrics 116, 2175–2188 (2018). https://doi.org/10.1007/s11192-018-2820-9*

*Haddaway NR, Collins AM, Coughlin D, Kirk S (2015) The Role of Google Scholar in Evidence Reviews and Its Applicability to Grey Literature Searching. PLoS ONE 10(9): e0138237. https://doi.org/10.1371/journal.pone.0138237*

Considering, the review process applied for collecting articles, as mentioned section 4.1, was used to pick up only the articles responding to the following search sequence, in the electronic databases, were considered: ("coastal flood*" OR "sea-level rise" OR "Storm surge" OR "Cyclonic storm" OR "Disaster risk reduction") AND ("Bangladesh" OR "Brahmaputra delta") AND ("*vulnerability" OR "human exposure" OR "coping*"). (see also answer to comment R1.2 and R1.7).

We propose to add this paragraph in the main text to clarify our review process (section 4.1):

"The initial search used inclusion criteria to be sure the first selected range of articles focused on studies about vulnerability to cyclonic floods in Bangladesh. A second selection of the first range of articles screened documents one by one to check if they explored vulnerability to cyclones and if they specifically did focus on areas placed in coastal Bangladesh, and not vulnerability to river flood for instance."

**R2.3** Although the literature review is well-organized and written, I would suggest that the author recheck a few recent scholarships, which may add new dimensions in vulnerability indicators.

Authors' response: Please, see also answers to comment R1.3 of the Referee#1.

Adnan et al. (2019) Have coastal embankments reduced flooding in Bangladesh? Science of The Total Environment, Volume 682, Pages 405-416

Authors' response: Adnan et al. (2019) analyzed the beneficial and harmful impacts of embankment construction on the territories. They focused on pluvial, fuvio-tidal and cyclonic floods and compared observed and modelled floods, with and without polders. As in Islam et al. (2019), this article focuses

on flooding, i.e. on the hazard, and not on the socio-spatial vulnerability to cyclonic flooding. This article was not therefore considered by our literature review process. (see also answer to comment R1.3).

**Islam Feroz et al. (2019) Flood risk assessment due to cyclone-induced dike breaching in coastal areas of Bangladesh, Nat. Hazards Earth Syst. Sci., 19, 353–368, 2019**

Authors' response: The main objective of Islam et al. (2019) is to estimate the exposed areas to storm surge flood of the polder 48 (Patuakhali district), using a hydrodynamic model under different scenarios in a context of climate change (tide, sea level, dike breach and cyclone landfall angle). This article focuses on flooding, i.e. on the hazard, and not on the population/territory vulnerability to cyclonic flooding. This article was not therefore considered by our literature review process. (see also answer to comment R1.3).

**Younus, M.A.F. (2017) An assessment of vulnerability and adaptation to cyclones through impact assessment guidelines: a bottom-up case study from Bangladesh coast. Nat Hazards 89, 1437–1459**

Authors' response: Younus et al. (2017) proposes an assessment of vulnerability based on a qualitative approach (by participatory rapid appraisal with semi-structured questions and groups of discussion) with the population of the Bawalkor village (Barguna district). Several high vulnerability issues are identified like seed-bed damage, primary occupation loss or culture fishpond loss. A second part of this study concerns adaptation issues (need to reconstruct the shelter, need relief from the interest of loan for 1 year after the disaster, need to reconstruct roads, etc). While interesting in its approach, this article does not identify key variables defining population/territory vulnerability to cyclonic flooding at the district level based on variables available in public databases. It was therefore not retained in our literature review process. (see also answer to comment R1.3).

**R2.4** Though the authors have mentioned based on their reviews that cyclonic storms are the major environmental threat in the coastal districts, also the tidal-surge induced floods needed to be considered, and the severity of vulnerability due to such tidal-surged induced inundations can change overall land use and livelihood pattern of the people. An example can be taken from cyclone Aila 2009, which also motivates people to shift from shrimp to rice cultivation which is absent in their discussion. Authors are suggested to revisit such land use transformation.

Authors' response: We agree with your comment that multiple physical processes are responsible for flooding at various spatial and temporal scale - including pluvial, fluvio-tidal, tidal, and cyclone-induced storm surge flooding (e.g., Haque et al. 2018). In this study, we focus only on the cyclone-induced storm surge flood hazard. However, it is important to note here that, in our modelling of the storm surge flooding probability, the tidal dynamics and riverine flow from Ganges, Brahmaputra, and Meghna are included. Please see our reply to point R1.4 concerning the flood complexity in coastal zone.

Concerning the land use transformation, according to our literature review, the transformation of land use following a disaster is rather a matter of the resilience of populations and their adaptive capacity, i.e. finding solutions to adapt to the new environmental constraints, as soil salinization (Hoque et al., 2017). Moreover, at the district scale, the land use is quite homogeneous and corresponds largely to agricultural areas.

*Hoque, S. F., C. H. Quinn, and S. Sallu. 2017. Resilience, political ecology, and well-being: an interdisciplinary approach to understanding social-ecological change in coastal Bangladesh. Ecology and Society 22(2):45. https://doi.org/10.5751/ES-09422-220245*

**R2.5** Temporal migration or translocal livelihood is one of the major aspects of the vulnerability constructs of the people living in this region. However, the author has not taken into consideration this

displacement indicator into their vulnerability index. Taking the displacement indicator of each district might result in different vulnerability indexes for major cities (i.e. districts).

Authors' response: Concerning the temporary, seasonal, or cyclical migration, from our literature review, only one article (Mallick and Vogt 2013) mentioned to taking a displacement indicator in the cyclonic flooding vulnerability estimation. However, this indicator was more mentioned as a post-disaster adaption strategy (e.g.; Younus and Sharna 2014; Sultana, 2010; Rahman et al. 2014; Mallick et al., 2017).

Even though there are historical patterns and systems of mobility of the Bangladeshi households, as strategy to cope with agricultural and environmental variability (Afsar, 2003; Call et al. 2017), the linkage between seasonal/temporal migrations and cyclonic flooding vulnerability is not straightforward. These processes are not a one-off phenomenon that occur in a specific place at a fixed date, making it difficult to measure. Kartiki (2013) showed that it is largely landless and poorest agricultural labourers who temporally/seasonally migrate. Seasonal migration, until six months, usually involves one or two male members of the household, migrating to nearby urban areas. If the temporal/seasonal migration is planned and remunerative this allows livelihood diversification of the household, asset accumulation and increases children school attendance (Kartiki (2013); Mobakar et al. 2002). Therefore, this temporal/seasonal displacement could be reduced indirectly the cyclonic flooding vulnerability. However, it is often the men, the heads of households, who migrate temporarily to find work. The rest of the household remains in place. The women are responsible for protecting and conserving goods and food, and in the event of a cyclone warning, they are reluctant to make the decision to evacuate. This means caring for all the people present (children and elderly) and taking them to a shelter, sometimes far away, leaving behind all their belongings and food reserves, which can then be stolen or destroyed. In this case a household without head household is indeed more vulnerable than a household where the household is complete. Thus, in this case, this temporal/seasonal displacement could increase indirectly the cyclonic flooding vulnerability.

*Afsar, R. Internal migration and the development nexus: the case of Bangladesh. In: Regional Conference on Migration, Development and Pro-Poor Policy Choices in Asia. 2003. p. 22-24.*

*Call, M. A., Gray, C., Yunus, M., & Emch, M. (2017). Disruption, not displacement: Environmental variability and temporary migration in Bangladesh. Global Environmental Change, 46, 157-165.*

*Kartiki, K. (2011). Climate change and migration: a case study from rural Bangladesh. Gender & Development, 19(1), 23-38.*

*Mobarak, A. M., & Reimão, M. E. (2020). Seasonal Poverty and Seasonal Migration in Asia. Asian Development Review, 37(1), 1-42.*

R2.6   Data collected from different years and sources raise the homogeneity in nature and the temporality of analysis. I propose to describe these as limitations.

Authors' response: The dataset periods used in this study vary between 2010 and 2018, but the major part comes from the BBS census of 2014. This is a strong limitation, but no public databases are available for more recent dates. For example, Das et al. (2020), Mullick et al. (2019) and Rabby et al. (2019) used the census of 2011.

Following this recommendation, we propose to add this text in the beginning of the new sub-section 6.2 Representativeness and quality of the data:

"A first limitation concerns the availability of dataset to construct the SSVI. Indeed, the dataset used for this study was produced at different dates: from 2010 for the poverty level, to 2018 for

paved/unpaved roads. However, the main information defining social vulnerability comes from the BBS Census of 2014, which gives some homogeneity to the description of the socio-demographic characteristics of the population."

R2.7    Besides the flood risks management is not very new in Bangladesh; it has a long history, including the polderization and adopted indigenous knowledge. It would be great if the author could add/ review how the flood risk management knowledge evolved in these coastal settings and justify why their study related to coastal flooding's socio-spatial vulnerability index is essential. Polderzation and the recent Delta Plan 2100 should be taken into consideration.

Authors' response: Risk management strategies are outlined in Section 3 Region study. They mention the reclamation policy since the 1960s, and the Cyclone Preparedness Program in the 1970s. However, nothing is specified for future years. We propose to mention the Delta Plan 2100 in the discussion section (see also answer to comment R1.11).

"Our results, for example, can help in the deployment of the Flood risk Management Strategies of the Delta Plan 2100 on the districts identified as failing on this dimension of vulnerability. Sub-strategies FR 1.1, 1.2 and 1.3 (Protection by development and improvements of embankments, barriers and water control structures (including ring dikes) for economic priority zones and major urban centres; Construct adaptive and flood-storm-surge resilient building; Adopt spatial planning and flood hazard zoning based on intensity of flood) for example correspond to the *Cyclone protection and exposure dimension* of the SSVI, for which Shariatpur and Jhalokati districts appear to be the most vulnerable (figure 4)."

*Delta Plan 2100,*
*http://plancomm.portal.gov.bd/sites/default/files/files/plancomm.portal.gov.bd/files/dc5b06a1_3a45_4ec7_951e_a9feac1ef783/B DP%202100%20Abridged%20Version%20English.pdf*

R2.8    The authors also address the cyclone shelter and evacuation process into the indicator lists. However, they do not mention how the community people decide the location of those cyclone shelter cum primary schools or any other community infrastructure and how the social elite controls the planning procedures.

Authors' response:  Following the recommendation of the Referee #2 and #1, we propose to add this paragraph in the discussion section (see also answer to comment R1.10).

"The shelter capacity is another example of variables that does not seems always representative of the situation on the field. It can be assumed that existence of cyclone shelters reduces people's exposure and therefore their vulnerability. However, presence of cyclone shelters in close proximity to homes, within a radius of 1 to 1.5km, does not mean they are useful and used. Like mentioned by Mallick et al. 2017, shelters are not placed on territories in a way that can accommodate as many people as possible and be accessible, but are rather near the supreme classes. These buildings are not always maintained and do not meet the requirements expected by the local society: Men and women are in a same mixed place, there are no women-only sanitary facilities, and there can be a certain amount of insecurity (Kulatunga et al., 2014; Saha, 2015)."

R2.9    These are significant short-comings along with the revision of the motivations of the study. I would suggest that authors specify why their contribution is novel despite having a larger body of literature on vulnerability assessment of coastal Bangladesh.

Authors' response: Please see our previous replies to points R2.1; and R1.1 and R1.11 concerning the novelty of the study.

---

## Author Response (AR2)

**MS No.: nhess-2021-8**

We thank the Editor and the reviewers for their constructive comments that have helped us to significantly improve our manuscript. Please find below a detailed response to the Referee#3 comments point by point. Following the Editor's suggestion we have been proofreading our manuscript by bilingual person. Changes and corrections appear in the track change document.

**Response to review comments by Anonymous Referee #3 (R3)**

**R3.1** Line 15, 48, 62: It appeared to me that the authors used the terminologies – "sensitivity" and "Susceptibility" interchangeably. These terminologies should not be used as jargon which is generally avoided. Each terminology has its specific definition and use.

**Answer:**

In the manuscript, the term "sensitivity" is used to refer to the degree to which a system is modified or affected, either adversely or beneficially,  by perturbations (following the definition of the IPCC AR4).

Then "Sensitivity [to cyclone flooding]" is properly used :

line 15 "sensitivity of delta inhabitants"

line 59 "the sensitivity of people "

line 62 "vulnerability and individual/group sensitivity"

line 68 "sensitivity of inhabitants"

line 514 "sensitivity of inhabitants"

line 646 "the sensitivity of delta inhabitants"

In the manuscript, the term "susceptibility" refers to the potential to suffer a loss as a result of perturbations.

Then "Susceptibility [to loss]" is also properly used :

line 48 "Vulnerability has many facets influencing each other and may also be measured by susceptibility to loss, on physical (natural and man-made environments), social, economic, institutional and systemic levels. "

**R3.2** Line 307: The author used the terminology "coastal wall" to refer to the coastal embankment which is not appropriate. Because the coastal wall is a specific type of structure that is different from the embankment.

**Answer:**

To avoid confusion, the term "coastal wall" has been replaced by "coastal embankment".

**R3.3** The methodology should explicitly mention the definition of vulnerability used in constructing the SSVI.

**Answer:**

We are not sure to understand the Referee's point. The definition of vulnerability used in our study is discussed in *Section 2: Overview about the vulnerability concept and its indicators* and in S*ection 4: Assessing vulnerability to cyclonic coastal flooding in Bangladesh from a literature review*.

We considered a cyclone flood event in this integrated and interdisciplinary framework, as the intersection between physical vulnerability and individual/group sensitivity, resulting into the concept of 'socio-spatial vulnerability' to cyclone flood (Forrest et al., 2020). We defined a new metric, called 'socio-spatial vulnerability index' (SSVI), as function of both the probability of the cyclone flood hazard and the sensitivity of inhabitants, plainly presented in *Section 5: A socio-spatial vulnerability index* and in *Figure 2*.

**R3.4** The author did not consider any indicators related to social capitals (e.g. network, relationship, etc,). It should be clearly explained with reasons that this SSVI did not include the social capital.

**Answer:**

The social capital is defined as the whole of the actual or potential resources which are linked to the possession of a durable network of more or less institutionalized relations of inter-knowledge and inter-recognition.

In our literature review, only three articles mention social capital: Islam et al, 2014, Islam and Wlakerden, 2014, and Ahsan and Warner, 2014. In these references, social capital is mentioned in the post-disaster phase as an important element of coping. However, in our approach, we have chosen to assess vulnerability without including coping as a component of the vulnerability, which we consider to be a separate and distinct area of research from the one of vulnerability.

Moreover, it is difficult to evaluate the social capital of individuals quantitatively without using surveys and, more generally, qualitative approaches, which is not the purpose of our research. This limit is mentioned in **section 6.3 Limits of the research**, lines 620 to 627.

**R3.5** The SSVI index considered the length of the embankment at each district as an indicator of cyclone protection (474). In lines 305-308 the author argued that these embankments are often in bad maintenance and there are some illegal activities that further weaken the embankment. Therefore, the length of the embankment may not be considered as an indicator of cyclone protection, especially the cyclone with 50 year return period without exploring the design of those indicators. Because it is also important to know whether those embankments were designed to withstand such an event.

**Answer:**

We remind here that the justification of each factor, used in the SSVI index computation, is drawn from our literature review process. Therefore, based on our literature review process, we hypothesized that the presence of dikes and embankments reduces the exposure of populations and territories located behind them. The variable *dikes and embankments* is cited in 39% of selected articles (i.e. 19 among 49 articles considered, see Appendix B) to define the vulnerability to cyclonic flooding (please see also our reply R1.9 to the Referee#1).

Moreover, although some dikes and embankments could be in poor condition, this information is currently not publicly available and is not accessible through databases neither.

In our hydrodynamic modelling we reported the water level with cyclonic surge with 50-year return period (for a climatology of 3600 synthetic cyclones) and not the "cyclone with 50 year return period " as mentioned by the referee. We have chosen the "50-year return period" because, the recent cyclones that made damages produced water level higher than 50-year return level. But in any case it is mean that the embankments are designed with 50-year return level, there is no connection to that. The embankments are what they are - as best as we could find from the available dataset.

Once again, based on our literature review, it seems reasonable to assume that they always play a role in protecting people against cyclonic flooding.

**R3.6** Moreover, the role of these embankments against storm surge is controversial. It is true that these embankments have been found effective somewhere against low storm surge but there is an issue of a false sense of security as well (Kibria & Khan, 2017; Parvin et al., 2019; Sadik et al., 2018; Shah Alam Khan, 2008). Further, the author should also mention that there are other important indicators/variables of vulnerability which they could not consider here, for example, access to cyclone shelter, evacuation behaviors, etc.

**Answer:**

We agree with this remark and in the *Discussion section*, we mention the limits of this variable (lines 582 to 598). Access to cyclone shelter and evacuation behavior are also mentioned and discussed, respectively, lines 599 to 610 and lines 295 to 299.

---

## Author Response (AR3)

**MS No.: nhess-2021-8**

Dear Editor,

Please find us kind of astonished to receive minor revisions on our work for the fourth time.
We would like stress the answers we already brought:
1) 19 Jan 2021 (Comments to the Editor – Minor revisions);
2) 01 May 2021( Referees #1 and #2 – Major revisions);
3) 20 Aug 2021( Referees #1 and #3 – Accepted as it, and Minor revisions).

However, as we are convinced that our work contributes to bring a paradigm shift, considering the integrated socio-spatial vulnerability, a new perspective to assess the vulnerability to cyclonic coastal flooding, and provides a framework for future research on this topic, we respond to these new comments with devotion, precision and objectivity, as it has been always the case.

In the following, we address you point-by-point all the answers and we will modify the manuscript accordingly.

***(a) How do you define coastal zone in Bangladesh? According to the policy document of Bangladesh, there are 19 coastal districts. This should be consistent.***

This remark is quite relevant, as the coastal zone can be defined according to several criteria. For example, Mullick et al., 2019 chose the same districts to conduct their study on coastal vulnerability. But to be more specific, we suggest adding this text, section 5 line 416:
"To identify places that are highly vulnerable to cyclonic flooding hazard, we focus on 16 coastal districts, the district being the spatial unit of reference (Fig. 1a). The 16 costal districts were selected following the classification given by the Program Development Office for Integrated Coastal Zone Management Plan (PDO-ICZMP, Uddin & Kaudstaal, 2003) which considered 12 "exposed" districts to the sea and/or lower estuaries (i.e Khulna, Satkhira, Barguna, Cox's Bazar, Bagerhat , Patuakhali, Pirojpur, Chittagong, Noakhali, Bhola, Lakshmipur and Feni). We added also to this selection 3 districts located on the mouth area of the GBM delta and cited in the literature review: Chandpur, Shariatpur and Barisal (Fig. 1a, see Appendix B.b). Moreover, the district of Jhalokati bounded by Barisal, Pirojpur and Barguna districts, is added to our selection because it is cited in the literature review in 5 studies."

Uddin, A. M. K., & Kaudstaal, R. (2003). Delineation of the coastal zone. Program Development Office for Integrated Coastal Zone Management Plant (PDO-ICZMP), Dhaka, 1–42.

***(b) The statistical database is available at sub-district (Upazila) or Union level. Why you used district level data, even data is available at lower administrative level. I would suggest to conduct the study at lower administrative level (i.e., Upazila). This will add more value for local decision makers.***

First and foremost, we agree with the editor about the need to work on upazila and union levels. It is exactly the focus we adopt in a further paper currently in preparation. Nonetheless,

working at the smaller administrative level would be a thorn in the side in the context of this study.

Using the district scale is related to the multifold aims of our research. First, one of the objectives was the spatial identification of hotspots of vulnerability to cyclonic flooding, intending both to identify the spatial variability linked to cyclonic flooding facing the district; and districts where the combined effects of multiple social, economic and environmental stressors are most prevalent regarding the cyclonic flooding.

Secondly, by using this territorial delimitation, it enables us **to zoom out** the research fieldworks made from 2007 on, in order to show which zones have been under the scope of science and other very exposed and vulnerable zones, almost forgotten from studies. While most of the analysed studies worked on local places, they did not give a whole picture to the reader about the work made in coastal Bangladesh. We detailed all the analysed sites studied by district, since an article could study many villages or upazila in the same district. The distribution of the studies on the map shows that two main districts on the west coast were over-studied (Shatkhira and Khulna) compared to the need of studying now the most vulnerable districts, situated in the mouth of the Delta. Our research consequently shows the gap between the existing scientific research and the geographical zones that should be prioritized by academics for further studies because of their socio-spatial vulnerability characteristics.

Moreover, the interest to study at the district scale is linked to the alert and the cyclone **warnings dissemination system in Bangladesh**. For instance, the community leaders in rural areas (the *samaj*) fill a special role in coastal districts, at informing the population about imminent hazards, via megaphones and by house-to-house contacts (Paul, 2009). They may also perform other duties such as maintaining cyclone shelters and assisting in evacuation procedures (Kulatunga et al., 2014), key-points among other factors of our defined vulnerability. In the post-disaster period, the *samaj* also contribute toward food, financial aid and other material support (Alam and Collins, 2010). It is therefore the scale to encompass key actors intervening during both the pre-disaster and the post-disaster phases of the cyclonic hazard.

Finally, our methodology was developed in order to support district assessments and provide suitable information for identifying districts where vulnerability to cyclonic flooding could be relatively high and for questioning the preparation of strategies for the elaboration of mitigation and adaptation policies in these hotspots.

***(c) When you described the results, I do not find any validation. At least you should support your results through secondary literature. Justification why certain area is more vulnerable than other?***

This point has been discussed previously as a response to a comment raised by the Referee#1 (see our reply R1.11). The definition of vulnerability varies greatly from one study to another, making a comparison difficult between several studies. However, the one made by Quader et al. (2017) is the most relevant to us in order to perform this exercise. Based on that R1.11 comment, we have already modified our discussion as follows lines 562-582 : " […]  comparing

vulnerability studies of coastal districts exposed to cyclonic flooding risk remains very difficult because the definition of the vulnerability concept varies greatly from one study to another. Quader et al. (2017) is certainly the most recent study offering a definition and an assessment of vulnerability, as well as the most relevant to our study. The results corroborate that the districts in the mouth of the Meghna (central coast) up to Chandpur and Shariatpur districts, as well as the Bagerhat and Cox's Bazar districts are highly vulnerable. The only notable difference is situated in Shatkhira district, which according to our study is vulnerable, while according to Quader et al. (2017) is very low vulnerable. In detail, the different dimensions of vulnerability are not described in the same way by the two studies. For example, accessibility to electricity is used to define demographic and basic facilities vulnerability in Quader et al. study, while it is used to define the vulnerability of infrastructure and housing in our SSVI. Similarly, disability is sometimes used to define household vulnerability (SSVI), and sometimes considered as a separate dimension of vulnerability (Quader et al., 2017). Moreover, we used a robust probabilistic cyclonic flood hazard map based on a dataset of 3600 statistically and physically consistent synthetic cyclone events (Emanuel et al., 2006; Khan et al. 2019). While Quader et al. (2017) used a low level of confidence cyclone hazard density interpolated map based on historical cyclone tracks (~160 events in 1877-2015). Although the level of detail provided by this union-wide study is significant, the closeness of the results is meaningful. However, social vulnerability is defined from 141 variables and a consistent workflow with several statistical methods. On the contrary, in our study, we have devoted a strong attention to the theoretical links between indicators and underlying vulnerability to cyclonic flooding, by conducting a strict literature review. Therefore, SSVI is computed from only 17 variables and a simple computation of index appears easier to understand its construction and to replicate by decision makers."

***Also, I would indicate key recommendations how vulnerability be reduced.***

Providing "key recommendations to reduced vulnerability" is valuable and interesting, but it is regrettably out of the scope of our study.

***(d) The most critical one: I do not find logical justification of choosing the approach: if you do not find any available studies on particular approach or indicators, you have not considered them. Then, what is the novelty of the study? The approach you considered might be applicable for review paper. For ensuring novelty, it should be other way around.***

We are not sure to correctly grasp the Editor's comment questioning upon *the approach*. As previously explained in the manuscript, the most common vulnerability indexes are based on inductive and deductive approaches. Inductive approaches use large sets of variables to build statistical models (e.g principal components analysis), that explain observed vulnerability through some indicating variables. Although a large number of variables may be useful for descriptive purposes, including non-influential variables in the index aggregation may decrease both explanatory power and user level of understanding. Deductive models can contain a few dozen variables, or less, which are normalized and aggregated to index, which could be separated into groups sharing the same underlying vulnerability dimension (e.g Fig2 : socioeconomic, household composition and disability...). This approach is the most common structure applied to vulnerability indices. Deductive approach, based on a data-driven mindset from expert knowledge (literature review) and parsimony, helps to identify and trace underlying

themes running through the data, opposite to inductive approach where the statistical models obfuscate underlying data. We consider that the literature review is a rich source to understand the main causes, translated as indicators of vulnerability, their relative importance and interactions.

It is important to keep in mind here that our work is not limited to the definition of a vulnerability index to cyclonic flooding but provides also a framework for future research on this topic based on:

- its entry by the territory seems more useful to us in order to help decision makers in their decision making and their strategic orientations in risk management,
- a global and deductive approach was given to the definition of vulnerability (and not only for each of its dimensions),
- a robust probabilistic cyclonic flood hazard map was made, based on a dataset of 3600 statistically and physically consistent synthetic cyclone events.
- and finally, the definition of vulnerability alone was elaborated, without the adaptive capacities, but including exposure.

This paper presents a strong scientific and conceptual contribution, providing a sound methodological design with a simple computation of index easier to understand and to replicate by decision makers. We are convinced that our work addresses a topic of significant community interest. This interest of the community is already marked by the number of consultations of the preprint (nhess metrics: 640 views since February 2021).

Therefore, it appears necessary to publish the article fast enough to offer those new findings to interested searchers, policy makers and IOs-NGOs. And by offering a conceptual bridge to and in between both natural hazard and social sciences upon the notion of vulnerability, we strongly believe the article will feed the reasoning for scientists working on vulnerability to cyclones in Bangladesh.

---

## Author Response (AR4)

**MS No.: nhess-2021-8**

Dear Editor,

We thank you to reconsider our revised manuscript for publication in NHESS.

Following your recommendations, we have modified the manuscript by addressing, point-by-point, all the issues raised. The proposed revisions in the updated manuscript are highlighted by yellow color.

We are looking forward to your assessment of the revised manuscript.

Yours sincerely,
Dr. N. Long, on behalf of the co-authors
* * *
**1. Representation of coastal area considering 19 districts of coastal area delineation map of Bangladesh.**

Added in the text, section 5 "A socio-spatial vulnerability index" line 415:

*"To identify places that are highly vulnerable to cyclonic flooding hazard, we focus on 16 coastal districts, the "district" being a spatial unit of reference (Fig. 1a). As widely discussed by the Program Development Office for Integrated Coastal Zone Management Plan (PDO-ICZMP, Uddin & Kaudstaal, 2003), the definition of a coastal zone is not simple and depends on selected criteria. According to the PDO-ICZMP classification, we have considered 12 "exposed" districts, i.e. ones adjacent to the sea and/or located in the lower estuaries (i.e Khulna, Satkhira, Barguna, Cox's Bazar, Bagerhat , Patuakhali, Pirojpur, Chittagong, Noakhali, Bhola, Lakshmipur and Feni). The interior coast districts (Jessore, Narail and Gopalganj) are less exposed to cyclone inundations, as the other districts facing the sea. Moreover, following the literature review (Fig. 1a, see Appendix B.b) and given the trajectories of major cyclones as Bhola, Gorky or Sidr, we considered that the districts in the mouth area, i.e Barisal, Shariatpur and Chandpur, are also highly exposed to cyclonic floodings. Finally, the district of Jhalokati, cited in the literature review and bounded by Barisal, Pirojpur and Barguna, is added to maintain a territorial coherence."*

*Uddin, A. M. K., & Kaudstaal, R. (2003). Delineation of the coastal zone. Program Development Office for Integrated Coastal Zone Management Plant (PDO-ICZMP), Dhaka, 1–42.*

**2. Significant variation of vulnerability can be found within a district (e.g., Khulna Sadar vs. Koyra Upazila), hence, analysing vulnerability at subdistrict level is essential.**

Added in the text, section 6.2 "**Representativeness and quality of the data at the district scale**" line 631:

*"The sub-district level case study presents advantages in data mining, the use of participatory methods and in capturing the fine complexity of vulnerability, but these studies are very local specific and therefore are difficult to compare with other regions. Moreover, the integration into a vulnerability index of different*

*local qualitative and quantitative data, which have their own logic of underlying scales and pattern, implies a loss of information and relevance (Fekete et al. 2010).*

*Here, using the "district" scale is related to the multifold aims of our research. One of the objectives was a spatial identification of hotspots of vulnerability to cyclonic flooding, intending both to identify the spatial variability linked to cyclonic flooding facing the district and the districts where the combined effects of multiple social, economic and environmental stressors are most prevalent regarding the cyclonic flooding. Therefore, the districts have been chosen as units of analysis for several motives: 1) districts are relatively homogeneous in size in comparison with sub-districts, municipalities or unions; 2) cyclonic hazard management are organized and supervised on the district level (e.g the alert and cyclone warnings dissemination system, maintaining cyclone shelters and assisting in evacuation procedures, Kulatunga et al., 2014); 3) a sufficient number of variables is freely available online from the Bangladesh Bureau of Statistics, 4) district unit can be more easily transposable and replicated on other delta regions (e.g applied in the 13 provinces of the Vietnamese Mekong Delta or in the 6 districts of the Ayeyarwady Delta); and 5) as administrative unit, the districts are easily understood by decision makers, planners and end-users.*

*Moreover, using this territorial delimitation enables us to zoom out the research fieldworks conducted from 2007, in order to show which zones have been under the scope of science and others very exposed and vulnerable but almost forgotten by researches. While most of the analyzed studies worked on local places, they did not give a whole picture to the reader about the work made in coastal Bangladesh. The distribution of the studies on the map (Fig.3) shows that two main districts on the west coast were over-studied (Shatkhira and Khulna) compared to the need of studying now the most vulnerable districts, situated in the mouth of the Delta."*

*Fekete, A., Damm, M., & Birkmann, J. (2010). Scales as a challenge for vulnerability assessment. Natural Hazards, 55(3), 729-747.*

**3a. Selection of vulnerability indicators based on scientific literature might not be adequate.**

Modified and added in the text section 5 "A socio-spatial vulnerability index" line 428:

[revised manuscript text omitted]

**4. Though title indicates, socio-spatial vulnerability, I found weak representation of social science here. e.g., no participatory approach, limited consideration of social indicators such as networks or collaboration pattern.**

The title of the manuscript is : *"Bangladesh's vulnerability to cyclonic coastal flooding"*.